# Gradient Descent with Linearly Correlated Noise: Theory and Applications to Differential Privacy

**Anastasia Koloskova**[*]
EPFL, Switzerland

**Ryan McKenna**
Google Research

**Zachary Charles**
Google Research

**Keith Rush**
Google Research

**Brendan McMahan**
Google Research

## Abstract

We study gradient descent under linearly correlated noise. Our work is motivated by recent practical methods for optimization with differential privacy (DP), such as DP-FTRL, which achieve strong performance in settings where privacy amplification techniques are infeasible (such as in federated learning). These methods inject privacy noise through a matrix factorization mechanism, making the noise linearly correlated over iterations. We propose a simplified setting that distills key facets of these methods and isolates the impact of linearly correlated noise. We analyze the behavior of gradient descent in this setting, for both convex and non-convex functions. Our analysis is demonstrably tighter than prior work and recovers multiple important special cases exactly (including anti-correlated perturbed gradient descent). We use our results to develop new, effective matrix factorizations for differentially private optimization, and highlight the benefits of these factorizations theoretically and empirically.

## 1 Introduction

Differential privacy (DP) is a critical framework for designing algorithms with provable statistical privacy guarantees. DP stochastic gradient descent (DP-SGD, Abadi et al. [1]) is particularly important for enabling private empirical risk minimization (ERM) of machine learning models. Many works have analyzed the convergence behavior of DP ERM methods, including DP-SGD [5, 16, 48, 8]. However, obtaining good privacy/utility trade-offs with DP-SGD can require excessively large batch sizes or privacy amplification techniques such as subsampling [4, 5, 55] and shuffling [15, 16]. In some applications, including cross-device federated learning, limited and device-controlled client availability can make sampling or shuffling infeasible [21]. Even outside of such applications, many implementations of DP-SGD do not properly use the Poisson subsampling scheme analyzed by Abadi et al. [1] for amplification, and instead use a single fixed permutation of the dataset [7].

Kairouz et al. [20] propose an alternative method, DP-FTRL, which can attain good privacy/utility trade-offs without amplification. Their key insight is that for SGD-style algorithms, the variance on *prefix sums* $\mathbf{g}_0 + \cdots + \mathbf{g}_t$, $t \in \{1, \ldots, T\}$ of gradients $\mathbf{g}_j$ is more important than the variance on individual gradients. By adding carefully tailored noise that is *linearly correlated* over iterations to the gradients, one can reduce the error on the prefix sums, at the cost of increased error on the individual gradients, for a fixed privacy budget. The DP-FTRL mechanism is competitive with or better than DP-SGD, even without relying on privacy amplification, and enabled McMahan and Thakurta [31] to train the first differentially private machine learning model on user data in a production setting.

---

[*]Work performed while doing an internship at Google Research. Correspondence to: Anastasia Koloskova <anastasia.koloskova@epfl.ch>, Ryan McKenna <mckennar@google.com>.

37th Conference on Neural Information Processing Systems (NeurIPS 2023).

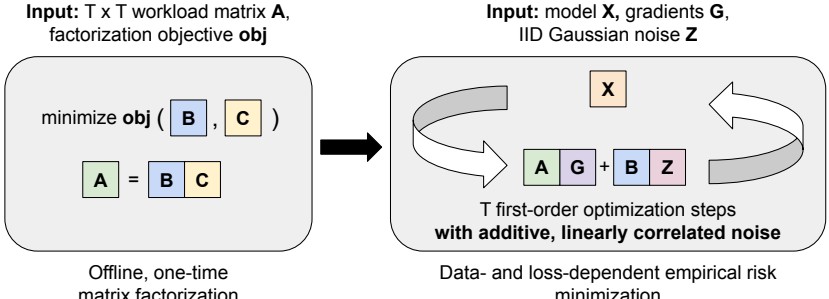

**Input:** T x T workload matrix **A**,
factorization objective **obj**

minimize **obj** ( **B** , **C** )

**A** = **B** **C**

Offline, one-time
matrix factorization

**Input:** model **X**, gradients **G**,
IID Gaussian noise **Z**

**X**

**A** **G** + **B** **Z**

T first-order optimization steps
**with additive, linearly correlated noise**

Data- and loss-dependent empirical risk
minimization

Figure 1: Two-stage MF-DP-FTRL workflow proposed by Denisov et al. [10]. The user selects a workload matrix $\mathbf{A}$ representing a desired first-order optimization method. Offline, the user finds a factorization $\mathbf{BC} = \mathbf{A}$, using an objective that balances ERM performance (as a function of $\mathbf{B}$) and privacy (as a function of $\mathbf{C}$). The user applies $\mathbf{A}$ to a downstream ERM task, but with linearly correlated additive noise governed by $\mathbf{B}$.

Denisov et al. [10], Choquette-Choo et al. [7] develop a refinement of DP-FTRL, MF-DP-FTRL, by formulating and solving an offline matrix factorization problem to find the "optimal" correlated noise structure under DP constraints. That is, for a fixed privacy level, they aim to find correlated noise structures that lead to improved optimization. A simplified diagram of their workflow is given in Fig. 1. However, (as we detail in Section 2) their offline factorization objective is based on an online convergence bound that is loose. This raises questions about whether there are factorization objectives that better capture convergence behavior of gradient descent algorithms with correlated noise.

In this paper we study this class of mechanisms more closely and provide a detailed analysis of linearly correlated noise from an optimization point of view. Our main contributions are as follows:

- We propose a novel stochastic optimization problem that extracts key facets of methods like (MF-)DP-FTRL, and which isolates the effects of linearly correlated noise on optimization.

- We derive convergence rates for gradient descent on smooth convex and non-convex functions in such settings that showcase the effect of linearly correlated noise and recover tight convergence rates in notable special cases. We use a novel proof technique that may be of independent interest.

- We use this theory to design a new objective for the offline matrix factorization workflow in Fig. 1. We show that solving this objective leads to MF-DP-FTRL mechanisms with improved convergence properties. We validate the mechanism empirically on a variety of datasets and tasks, matching or outperforming prior methods.

## 1.1 Related Work

**Matrix mechanisms for differential privacy.** Our work is closely related to differentially private optimization using matrix mechanisms [26]. Historically, such mechanisms were applied to linear statistical queries [25, 29, 14, 18]. Denisov et al. [10] and Choquette-Choo et al. [7] extended these mechanisms to the adaptive streaming setting, allowing their application to optimization with DP. Denisov et al. [10] show that this framework (MF-DP-FTRL) subsumes and improves the DP-FTRL algorithm [20]. Both DP-FTRL and MF-DP-FTRL improve privacy guarantees relative to DP-SGD [1] without amplification, and can be combined with techniques such as momentum for improved utility [46]. The aforementioned work focuses on methods for computing factorizations, privacy properties, and empirics. Our work studies the analytic relationship between the correlated noise induced by the MF-DP-FTRL framework and the downstream effect on optimization performance.

**SGD with correlated noise.** Stochastic noise in optimization arises in a variety of ways, including mini-batching [9] and explicit noise injection [11, 54, 19]. While most analyses of SGD assume this noise is independent across iterates, some work considers correlated noise. For example, shuffle SGD involves correlated noise due to sampling without replacement [33, 53]. Lucchi et al. [27]

use correlated Brownian motion to improve SGD's ability to explore the loss landscape. Recently, Orvieto et al. [37, 38] investigated anti-correlated noise as a way to impose regularization and improve generalization. We consider a linearly correlated noise model, and analyze its impact on SGD's convergence to critical points.

**SGD with biased noise.** Many algorithms can be viewed as SGD with structured but potentially biased noise, including SGD with (biased) compression [44, 17], delayed SGD [28, 12], local SGD [42], federated learning methods [22, 52, 34, 36], decentralized optimization methods [50, 23], and many others. Convergence analyses for such methods often use techniques like perturbed iterate analysis [28]. Correlated gradient noise also biases the gradient updates. However, as we show in Section 4, directly applying such techniques to linearly correlated noise does not lead to tight convergence guarantees.

## 2 Background

In this work, we focus on an empirical risk minimization (ERM) problem of the form

$$\min_{\mathbf{x} \in \mathbb{R}^d} \left[ f(\mathbf{x}) = \frac{1}{n} \sum_{i=1}^{n} l(\mathbf{x}, \xi_i) \right], \tag{1}$$

where $l(\mathbf{x}, \xi_i)$ is the loss of a model $\mathbf{x}$ on a data point $\xi_i$, and $n$ is the training set size. We would like to solve (1) while guaranteeing some form of privacy for the training set. We focus on *differential privacy* (DP, [13]), a widely-used standard for anonymous data release. DP guarantees statistical difficulty in distinguishing whether or not a particular unit's data served as an input to a given algorithm, based on the algorithm's output. This protected unit may represent a single training example or a semantically higher-level unit like the entirety of a user's data.

While there are many methods for solving (1), we will follow Denisov et al. [10], Choquette-Choo et al. [7] and restrict to first-order algorithms $\mathfrak{A}$ that linearly combine (stochastic) gradients. Each algorithm $\mathcal{A} \in \mathfrak{A}$ is parameterized by a learning rate $\gamma > 0$, a number of steps $T > 0$, and scalars $\{a_{tj}\}_{1 \leq j \leq t \leq T}$. Given a starting point $\mathbf{x}_0$, $\mathcal{A}$ produces iterates $\mathbf{x}_t \in \mathbb{R}^d$ given by

$$\mathbf{x}_{t+1} = \mathbf{x}_0 - \gamma \mathcal{A}_t(\mathbf{g}_1, \ldots, \mathbf{g}_t) \qquad \mathcal{A}_t(\mathbf{g}_1, \ldots, \mathbf{g}_t) = \sum_{j=1}^{t} a_{tj} \mathbf{g}_j$$

where $\mathbf{g}_t$ is a (mini-batch) gradient of $f$ computed at $\mathbf{x}_t$. This class encompasses a variety of first-order algorithms, including SGD [40], SGD with momentum [39, 35], and delayed SGD [2]. This class also captures algorithms that use learning rate scheduling, so long as the schedule is independent of the gradient values. We re-write the output of $\mathcal{A}$ in matrix notation by defining:

$$\mathbf{X} = [\mathbf{x}_1, \ldots, \mathbf{x}_T]^\top \in \mathbb{R}^{T \times d}, \ \mathbf{X}_0 = [\mathbf{x}_0, \ldots, \mathbf{x}_0]^\top \in \mathbb{R}^{T \times d}$$
$$\mathbf{G} = [\mathbf{g}_1, \ldots, \mathbf{g}_T]^\top \in \mathbb{R}^{T \times d}, \ \mathbf{A} = [a_{ij}]_{1 \leq i,j \leq T} \in \mathbb{R}^{T \times T}$$

Here $\mathbf{A}$ is the *workload matrix* representing $\mathcal{A}$. At iteration $t$, $\mathcal{A}$ can only use the current and previous gradients, so $a_{tj} = 0$ for $j > t$ (ie. $\mathbf{A}$ is lower-triangular). In this notation, the iterates of $\mathcal{A}$ satisfy

$$\mathbf{X} = \mathbf{X}_0 - \gamma \mathbf{A} \mathbf{G}. \tag{2}$$

**Example 2.1** (SGD). Define the *prefix-sum* matrix $\mathbf{S} \in \mathbb{R}^{T \times T}$ as the all-ones lower-triangular matrix. If $\mathbf{A} = \mathbf{S}$, then (2) is simply SGD with learning rate $\gamma$. As discussed by Denisov et al. [10, Section 4], we also recover SGD with momentum using an appropriate transformation $\mathbf{S}'$ of $\mathbf{S}$.

### 2.1 Matrix Factorization and Privacy Mechanisms

In order to make the output of (2) differentially private, we typically need to clip the gradients and add noise. Let $\overline{\mathbf{G}}$ denote the matrix whose rows (gradients) have been clipped to some $\ell_2$ threshold $\alpha$. Let $\mathbf{Z} \in \mathbb{R}^{T \times d}$ be a matrix with entries drawn independently from $\mathcal{N}(0, \varsigma^2/d)$. The well-known DP-SGD algorithm [1] adds this noise to each clipped gradient, so that

$$\mathbf{X} = \mathbf{X}_0 - \gamma \mathbf{A}(\overline{\mathbf{G}} + \mathbf{Z}). \tag{3}$$

For consistency, we consider (2) to be the special case of (3) where $\mathbf{Z} = \mathbf{0}$ and $\alpha = \infty$. The variance $\varsigma^2$ depends on the clipping threshold $\alpha$ and desired $(\varepsilon, \delta)$ privacy we aim to achieve [1].

To derive algorithms with improved DP guarantees, Denisov et al. [10] add the noise $\mathbf{Z}$ to a factorized version of $\mathbf{A}$. For a factorization $\mathbf{A} = \mathbf{BC}$ with $\mathbf{B}, \mathbf{C} \in \mathbb{R}^{T \times T}$, we add noise to the iterates via:

$$\mathbf{X} = \mathbf{X}_0 - \gamma \mathbf{B} \left( \mathbf{C}\overline{\mathbf{G}} + \text{sens}(\mathbf{C})\mathbf{Z} \right) \equiv \mathbf{X}_0 - \gamma \left( \mathbf{A}\overline{\mathbf{G}} + \text{sens}(\mathbf{C})\mathbf{BZ} \right). \tag{4}$$

Here, $\text{sens}(\mathbf{C})$ is a number representing the sensitivity of the mapping $\overline{\mathbf{G}} \mapsto \mathbf{C}\overline{\mathbf{G}}$ to "adjacent" input changes. We note that the sensitivity changes depending on the notion of adjacency. In single-epoch settings, two input matrices are adjacent if they differ by a single row [10], so the sensitivity function is $\text{sens}(\mathbf{C}) := \max_{i \in \{1, \dots, T\}} \|\mathbf{C}_{[:,i]}\|_2$, i.e. the maximum $\ell_2$-squared column norm of $\mathbf{C}$. For details and extensions to multiple epochs, see [7].

If the variance of entries of $\mathbf{Z}$ is fixed to some value $\varsigma^2/d$, then for all the possible factorizations $\mathbf{A} = \mathbf{BC}$ in (4) have exactly same privacy guarantees, depending only on $\varsigma$. It will also be convenient to define $\sigma = \text{sens}(\mathbf{C})\varsigma$ as the 'effective' variance of $\mathbf{Z}$ after re-scaling by the sensitivity. Note that for a fixed $\sigma$, the privacy guarantees of (4) might be different depending on the sensitivity.

The factorization $\mathbf{B} = \mathbf{A}, \mathbf{C} = \mathbf{I}$ recovers DP-SGD (3), but factorizations with better privacy-utility trade-offs may exist. The formulation of Eq. (4) transfers the linear optimization algorithm (2) into the setting of the matrix mechanism [26], a well-studied family of mechanisms in differential privacy. Denisov et al. [10], Choquette-Choo et al. [7] show that the mechanism in Eq. (4) provides a DP guarantee equivalent to a single application of the Gaussian mechanism, which can be computed tightly using numerical accounting techniques [49, 24].

**Finding good factorizations.** Intuitively, a factorization $\mathbf{A} = \mathbf{BC}$ is good if $\text{sens}(\mathbf{C})$ is small and the added noise $\mathbf{BZ}$ does not significantly degrade the convergence of (4). In order to quantify the effect of this added correlated noise on optimization, Denisov et al. [10] derive an online regret bound for (4) in the convex case against an adaptive adversary. Translating this via online-to-batch convergence to the stochastic setting, the iterates $\mathbf{x}_t$ satisfy

$$\frac{1}{T+1} \sum_{t=0}^{T} \mathbb{E}\left[ f(\mathbf{x}_t) - f^\star \right] \leq \mathcal{O}\left( \frac{\|\mathbf{x}_0 - \mathbf{x}^\star\|^2}{\gamma T} + \gamma \tilde{L}^2 + \gamma \varsigma \tilde{L} \frac{\text{sens}(\mathbf{C}) \|\mathbf{B}\|_F}{\sqrt{T}} \right) \tag{5}$$

where $\tilde{L}$ is the Lipshitz constant of $f$. Denisov et al. [10] therefore use $\text{sens}(\mathbf{C}) \|\mathbf{B}\|_F$ as a proxy for the impact of the factorized noise scheme on convergence. To find factorizations with good convergence properties, Denisov et al. [10], Choquette-Choo et al. [7] minimize $\text{sens}(\mathbf{C}) \|\mathbf{B}\|_F$ subject to the constraint $\mathbf{A} = \mathbf{BC}$, which is equivalent to the following objective:

*Problem* 2.2 (Minimal-Norm Matrix Factorization). Given a lower triangular matrix $\mathbf{A} \in \mathbb{R}^{T \times T}$, define $\text{OPT}_F(\mathbf{A}) = (\mathbf{B}, \mathbf{C})$, where $\mathbf{B}, \mathbf{C} \in \mathbb{R}^{T \times T}$ solve the following optimization problem.

$$\min_{\mathbf{B}, \mathbf{C}} \|\mathbf{B}\|_F^2 \quad \text{such that} \quad \mathbf{BC} = \mathbf{A}, \; \text{sens}(\mathbf{C}) = 1. \tag{6}$$

Eq. (6) is well-studied in the privacy literature and can be solved with a variety of numerical optimization algorithms [51, 30, 10, 7]. We also note that Denisov et al. [10] show that without loss of generality, we can assume $\mathbf{B}$ and $\mathbf{C}$ are lower triangular.

**Finding improved factorizations.** We argue that (5) is pessimistic in stochastic settings. For SGD (when $\mathbf{B} = \mathbf{A}$), the last term in (5) is $\mathcal{O}(\gamma \text{sens}(\mathbf{C})\varsigma \tilde{L} \sqrt{T})$, which diverges with $T$ for a constant stepsize. However, under the same assumptions as in [10], SGD with constant stepsize actually achieves a faster rate of $\mathcal{O}(\gamma \text{sens}(\mathbf{C})\varsigma \tilde{L})$ (see [41]).

In this paper, we turn our attention to the *smooth functions* in order to focus on non-convex functions. We show in Appendix A, there are matrices $\mathbf{B}_1, \mathbf{B}_2$ such that $\text{sens}(\mathbf{C}_1)\|\mathbf{B}_1\|_F = \text{sens}(\mathbf{C}_2)\|\mathbf{B}_2\|_F$, but Eq. (4) diverges with $\mathbf{B}_1$ and converges with $\mathbf{B}_2$, therefore showing that Frobenius norm is not the right measure in the smooth case as well.

This begs the question of whether there are objectives that better capture the impact of the noise injected in (4) on convergence. To answer this, we derive a bound that can exhibit better dependence on $\mathbf{B}$ to design better factorizations for differentially private optimization.

## 3 Problem Formulation

To study the effect of the noise $\mathbf{BZ}$ on optimization, we analyze a slightly simplified objective that omits parts of (4) not directly related to linear noise correlation. We do this as follows:

- (I) We assume that each $\mathbf{g}_t$ is the true gradient at the point $\mathbf{x}_t$, i.e. $\mathbf{g}_t = \nabla f(\mathbf{x}_t)$.
- (II) We omit gradient clipping from our analysis. Alternatively, we can view this as setting the clipping threshold $\alpha = \infty$ so that $\overline{\mathbf{G}} = \mathbf{G}$ in (4).
- (III) We restrict the class $\mathcal{A}$ to SGD-type algorithms where $\mathbf{A} = \mathbf{S}$, as in Example 2.1.

We impose (I) for simplicity of presentation. Our results can be extended to stochastic gradients in a direct fashion. Restriction (II) is also for simplicity. First, clipping is not directly applied to the noise $\mathbf{BZ}$. Second, for bounded domains or Lipschitz $f$, our analysis still holds with clipping. Last, practical DP methods often use adaptive clipping [45] instead of fixed clipping. We are not aware of convergence analyses for such schemes. We impose (III) in order to limit the class of algorithms $\mathcal{A}$ to a well-understood subclass. The convergence properties of (2) for general matrices $\mathbf{A}$ are not well-understood even when there is no noise ($\mathbf{Z} = \mathbf{0}$). As we discuss in Section 4, even with these simplifications, the effect of $\mathbf{BZ}$ is not well-understood.

Due to (III), we study factorizations $\mathbf{BC}$ of the matrix $\mathbf{A} = \mathbf{S}$, as in Example 2.1. Then, (4) becomes

$$\mathbf{X} = \mathbf{X}_0 - \gamma \left( \mathbf{SG} + \mathrm{sens}(\mathbf{C})\mathbf{BZ} \right). \tag{7}$$

In vector notation, for $\mathbf{b}_0 = \mathbf{0}$ and $\mathbf{B} = [\mathbf{b}_1, \dots \mathbf{b}_T]^\top$,

$$\mathbf{x}_{t+1} = \mathbf{x}_t - \gamma \left[ \nabla f(\mathbf{x}_t) + (\mathbf{b}_{t+1} - \mathbf{b}_t)^\top \mathbf{Z} \right], \tag{8}$$

where for simplicity of presentation, we re-scaled the noise $\mathbf{Z}$ by the sensitivity, $\sigma^2 = \mathrm{sens}^2(\mathbf{C})\zeta^2$. We now discuss several noteworthy special cases of (8).

**Example 3.1** (PGD). If $\mathbf{B} = \mathbf{S}$ (see Example 2.1) we recover SGD with uncorrelated additive noise, also known as *perturbed gradient descent* (PGD), where

$$\mathbf{x}_{t+1} = \mathbf{x}_t - \gamma \left[ \nabla f(\mathbf{x}_t) + \mathbf{z}_{t+1} \right]. \tag{9}$$

The convergence rate of SGD (and therefore PGD) is well-understood in the optimization literature (e.g. see Bubeck [6, Section 6]).

**Example 3.2** (Anti-PGD). By setting $\mathbf{B} = \mathbf{I}$, we get an algorithm that at every iteration adds an independent noise vector $\mathbf{z}_{t+1}$ and subtracts the previously added noise $\mathbf{z}_t$:

$$\mathbf{x}_{t+1} = \mathbf{x}_t - \gamma \left[ \nabla f(\mathbf{x}_t) + \mathbf{z}_{t+1} - \mathbf{z}_t \right], \quad \mathbf{z}_0 = \mathbf{0} \tag{10}$$

Intuitively, this removes some of the noise added in the prior round. This is (up to a learning rate factor) the *anti-correlated perturbed gradient descent* (Anti-PGD) method proposed by Orvieto et al. [37], who study its generalization properties. Anti-PGD is also equivalent to SGD with randomized-smoothing [11]. The equivalence follows from defining $\tilde{\mathbf{x}}_t = \mathbf{x}_t + \gamma \mathbf{z}_t$ and rewriting (10) as

$$\tilde{\mathbf{x}}_{t+1} = \tilde{\mathbf{x}}_t - \gamma \nabla f(\tilde{\mathbf{x}}_t - \gamma \mathbf{z}_t).$$

While randomized smoothing algorithm is popular for non-smooth optimization, Vardhan and Stich [47] analyze its convergence properties in the smooth non-convex setting.

**Example 3.3** (Tree Aggregation DP-FTRL). For $k \geq 1$ and $t = 2^{k-1}$, define $\mathbf{H}_k \in \mathbb{R}^{(2^k - 1) \times t}$ recursively as follows:

$$\mathbf{H}_1 = (1), \quad \mathbf{H}_{k+1} = \begin{pmatrix} \mathbf{H}_k & \mathbf{0} \\ \mathbf{0} & \mathbf{H}_k \\ \mathbf{1} & \mathbf{1} \end{pmatrix}$$

where $\mathbf{1}$ above represents an all-ones row of appropriate width. For $T = 2^{k-1}$, if $\mathbf{C} = \mathbf{H}_k$ and $\mathbf{B} = \mathbf{S}\mathbf{C}_k^\dagger$ where $\mathbf{C}_k^\dagger$ denotes a carefully chosen right pseudo-inverse of $\mathbf{C}$, then we recover the same noise matrix $\mathbf{B}$ as in the DP-FTRL algorithm with either the online or full Honaker estimator (depending on the choice of $\mathbf{C}^\dagger$) as in [20, 10]. Note that $\mathbf{B}, \mathbf{C}$ are not square. This can be remedied by appropriately projecting onto $\mathbb{R}^T$. See Choquette-Choo et al. [7, Appendix D.3] for details.

# 4 Deriving Tighter Convergence Rates

We would like convergence rates for (7) that apply to any factorization and yield tight convergence rates for notable special cases. We pay special attention to PGD (Example 3.1) and Anti-PGD (Example 3.2), as they represent extremes in the space of factorizations ($\mathbf{S} = \mathbf{SI}$ and $\mathbf{S} = \mathbf{IS}$, respectively). As we will show, it is possible to use existing theoretical tools to derive tight convergence rates for both, *but not simultaneously*.

Below, we discuss ways to derive tight rates for PGD and Anti-PGD, and how these rates involve incompatible analyses. We then develop a novel analytic framework involving *restart iterates* that allows us to analyze both methods simultaneously, as well as (7) for general factorizations. We start by formally stating our assumptions. For simplicity of presentation, we re-scale the noise $\mathbf{Z}$ by the sensitivity of $\mathbf{C}$, i.e. $\sigma^2 = \text{sens}^2(\mathbf{C})\zeta^2$; we will suppress the $\mathbf{C}$ dependence of $\sigma$.

**Assumption 4.1** (Noise). The rows $\mathbf{z}_1, \ldots, \mathbf{z}_T$ of the noise matrix $\mathbf{Z}$ are independent random vectors such that $\forall t$, $\mathbb{E}[\mathbf{z}_t] = \mathbf{0}$ and $\mathbb{E}\|\mathbf{z}_t\|^2 \leq \sigma^2$.

We do not assume $\tilde{L}$-Lipshitzness in our results, but we do assume $L$-smoothness. This is a relatively standard assumption in optimization literature [6].

**Assumption 4.2** (L-smoothness). The function $f : \mathbb{R}^d \to \mathbb{R}$ is differentiable, and there exists $L > 0$ such that for all $x, y \in \mathbb{R}^d$, $\|\nabla f(\mathbf{x}) - \nabla f(\mathbf{y})\| \leq L\|\mathbf{x} - \mathbf{y}\|$.

For *some* of the results we will assume convexity.

**Assumption 4.3** (Convexity). The function $f : \mathbb{R}^d \to \mathbb{R}$ is convex, i.e. $\forall \mathbf{x}, \mathbf{y} \in \mathbb{R}^d$, $f(\mathbf{x}) - f(\mathbf{y}) \leq \langle \nabla f(\mathbf{x}), \mathbf{x} - \mathbf{y} \rangle$. When assuming convexity, we also assume the infimum of $f$ is achieved in $\mathbb{R}^d$.

## 4.1 Convergence Rates for PGD and Anti-PGD

In this section we discuss the (distinct) convergence analyses of PGD and Anti-PGD, and the suboptimal results derived by trying to apply the proof technique for one to the other. We focus on the convex setting for brevity, though these analyses can be directly extended to the non-convex setting.

**PGD.** The convergence of PGD (Example 3.1) is well-understood since it is a special case of SGD. One can show the following.

**Proposition 4.4** (Adapted from Dekel et al. [9, Theorem 1]). *Under Assumptions 4.1, 4.2 and 4.3, if* $\mathbf{B} = \mathbf{S}$ *and* $\gamma < 1/2L$*, then the output of* (7) *satisfies*

$$\sum_{t=0}^{T} \frac{\mathbb{E}\left[f(\mathbf{x}_t) - f^\star\right]}{T + 1} \leq \mathcal{O}\left(\frac{\|\mathbf{x}_0 - \mathbf{x}^\star\|^2}{\gamma T} + \gamma \sigma^2\right). \tag{11}$$

The proof follows from combining the update (9), standard facts about convex functions, and the fact that $\gamma < 1/2L$, to get the inequality

$$\mathbb{E}_t \|\mathbf{x}_{t+1} - \mathbf{x}^\star\|^2 \leq \|\mathbf{x}_t - \mathbf{x}^\star\|^2 - \gamma\left(f(\mathbf{x}_t) - f^\star\right) + \gamma^2 \sigma^2.$$

It is left to average over iterations $0 \leq t \leq T$.

**Anti-PGD.** For Anti-PGD (Example 3.2), one can show the following.

**Proposition 4.5.** *Under Assumptions 4.1, 4.2 and 4.3, if* $\mathbf{B} = \mathbf{I}$ *and* $\gamma < 1/2L$*, then the output of* (7) *satisfies*

$$\sum_{t=0}^{T} \frac{\mathbb{E}\left[f(\mathbf{x}_t) - f^\star\right]}{T + 1} \leq \mathcal{O}\left(\frac{\|\mathbf{x}_0 - \mathbf{x}^\star\|^2}{\gamma T} + L\gamma^2 \sigma^2\right) \tag{12}$$

Since $L\gamma < 1/2$, the RHS of (12) is strictly smaller than the RHS of (11). While this result may be known, we were unable to find a reference, so we provide a complete proof in Appendix D. The proof utilizes perturbed iterate analysis [28]. We define a *virtual sequence* $\{\tilde{\mathbf{x}}_t\}_{t=0}^{T}$ as follows:

$$\tilde{\mathbf{x}}_{t+1} = \tilde{\mathbf{x}}_t - \gamma \nabla f(\mathbf{x}_t), \qquad\qquad \tilde{\mathbf{x}}_0 = \mathbf{x}_0 \tag{13}$$

The $\tilde{\mathbf{x}}_t$ are the iterates of (7) when $\mathbf{Z} = \mathbf{0}$. We can then prove the following descent inequality:

$$\|\tilde{\mathbf{x}}_{t+1} - \mathbf{x}^\star\|^2 \leq \|\tilde{\mathbf{x}}_t - \mathbf{x}^\star\|^2 - \frac{\gamma}{2}\left(f(\mathbf{x}_t) - f^\star\right) + 2L\gamma\|\tilde{\mathbf{x}}_t - \mathbf{x}_t\|^2.$$

Because of the anti-correlation in (10), the virtual iterates $\tilde{\mathbf{x}}_t$ are close to the real iterates $\mathbf{x}_t$, as $\mathbf{x}_t - \tilde{\mathbf{x}}_t = \gamma\mathbf{z}_t$. Averaging over $t$, we recover (12). See Appendix D for details.

**Tightness.** The noise terms (those terms involving $\sigma^2$) in (11), (12) are both tight. We show this in Appendix E on the objective $f(\mathbf{x}) = (L/2)\|\mathbf{x}\|^2$.

**Difficulties in a unified analysis.** The proof techniques for PGD and Anti-PGD above are notably different, and as we explain in Appendix F, do not lead to favorable results when trying to use one of the two strategies to analyze both.

## 4.2   Main Results and Analytic Techniques

To unify the proof techniques above, we use a modified virtual sequence with *restart iterations*. For a parameter $\tau = \tilde{\Theta}(1/L\gamma)$ (throughout, $\tilde{\mathcal{O}}$ and $\tilde{\Theta}$ hide poly-logarithmic factors), we define

$$\begin{aligned} \tilde{\mathbf{x}}_{t+1} &= \tilde{\mathbf{x}}_t - \gamma\nabla f(\mathbf{x}_t) & \text{if } t+1 \neq 0 \bmod \tau \\ \tilde{\mathbf{x}}_{t+1} &= \mathbf{x}_{t+1} & \text{if } t+1 = 0 \bmod \tau. \end{aligned} \quad (14)$$

Similar to the virtual sequence in (13), $\tilde{\mathbf{x}}_t$ incorporates only deterministic gradients $\nabla f(\mathbf{x}_t)$. However, every $\tau$ iterations we reset $\tilde{\mathbf{x}}_t$ to the real iterate $\mathbf{x}_t$. This allows us to control the divergence between the virtual sequence and the real sequence (enabling a tight analysis of PGD), while still capturing the convergence benefits of anti-correlated noise (enabling a tight analysis of Anti-PGD).

The parameter $\tau$ is independent of $\mathbf{B}$, and depends only on the geometry of $f$ and the stepsize $\gamma$. Using this machinery, we can prove convergence rates of (7) for *any* factorization $\mathbf{S} = \mathbf{BC}$. These rates involve $\ell_2$ distances between the rows $\mathbf{b}_t$ of the matrix $\mathbf{B}$ (where $\mathbf{b}_0 = \mathbf{0}$ for convenience).

**Theorem 4.6** (non-convex). *Suppose Assumptions 4.1 and 4.2 hold, $\gamma \leq 1/4L$, and $\tau = 1/\gamma L$. Then* (7) *produces iterates whose average error* $(T+1)^{-1}\sum_{t=0}^{T}\mathbb{E}\|\nabla f(\mathbf{x}_t)\|^2$ *is upper bounded by*

$$\mathcal{O}\left(\frac{(f(\mathbf{x}_0) - f^\star)}{\gamma T} + \frac{\sigma^2}{T\tau} \times \left[\frac{1}{\tau}\sum_{t=1}^{T}\left\|\mathbf{b}_t - \mathbf{b}_{\lfloor\frac{t}{\tau}\rfloor\tau}\right\|^2 + \sum_{\substack{1 \leq t \leq T \\ t=0 \bmod \tau}}\|\mathbf{b}_t - \mathbf{b}_{t-\tau}\|^2\right]\right).$$

**Theorem 4.7** (convex). *Under Assumptions 4.1, 4.2, and 4.3, if $\gamma \leq 1/4L$ and $\tau = \tilde{\Theta}(1/\gamma L)$, then* (7) *produces iterates with average error* $(T+1)^{-1}\sum_{t=0}^{T}\mathbb{E}\left[f(\mathbf{x}_t) - f^\star\right]$ *upper bounded by*

$$\tilde{\mathcal{O}}\left(\frac{\|\mathbf{x}_0 - \mathbf{x}^\star\|^2}{\gamma T} + \frac{\sigma^2}{TL\tau} \times \left[\frac{1}{\tau}\sum_{t=1}^{T}\left\|\mathbf{b}_t - \mathbf{b}_{\lfloor\frac{t}{\tau}\rfloor\tau}\right\|^2 + \sum_{\substack{1 \leq t \leq T \\ t=0 \bmod \tau}}\|\mathbf{b}_t - \mathbf{b}_{t-\tau}\|^2 + \left\|\mathbf{b}_{\lfloor\frac{T}{\tau}\rfloor\tau}\right\|^2\right]\right).$$

We give complete proofs in Appendix C. These convergence rates consist of two terms: The first term states how fast the function would converge in the absence of the noise. The second term, the *noise term*, is the focus of our paper, as it shows how the correlated noise $\mathbf{BZ}$ affects convergence.

These rates involve only differences of rows of $\mathbf{B}$ that are at most $\tau$ iterations apart. Intuitively, $\tau$ is a coarse indicator of whether an iterate $\mathbf{x}_t$ is still sensitive to the noise injected at an iteration $t' < t$. If $t > t' + \tau$, then changes in the noise added at step $t$ are effectively uncorrelated to iteration $t'$. As we detail in Appendix, applying Theorem 4.7 to the special cases in Examples 3.1, 3.2 recovers their tight convergence rates in (11), (12) correspondingly.

## 5   Finding Better Factorizations

We now draw on our results in Section 4 to develop better mechanisms for the MF-DP-FTRL framework. We modify the objective underlying the offline matrix factorization problem during the first stage of the MF-DP-FTRL workflow (Fig. 1). Specifically, observe that the noise term in Theorems 4.6 and 4.7 can be rewritten in matrix notation (up to multiplicative constants) as

$$\|\mathbf{\Lambda}_\tau\mathbf{B}\|_F^2 = \sum_{t=1}^{T}\left\|\boldsymbol{\lambda}_t^\top\mathbf{B}\right\|^2 = \sum_{\substack{1 \leq t \leq T \\ t=0 \bmod \tau}}\|\mathbf{b}_t - \mathbf{b}_{t-\tau}\|^2 + \sum_{\substack{1 \leq t \leq T \\ t \neq 0 \bmod \tau}}\left\|\frac{1}{\sqrt{\tau}}\left(\mathbf{b}_t - \mathbf{b}_{\lfloor\frac{t}{\tau}\rfloor\tau}\right)\right\|^2 \quad (15)$$

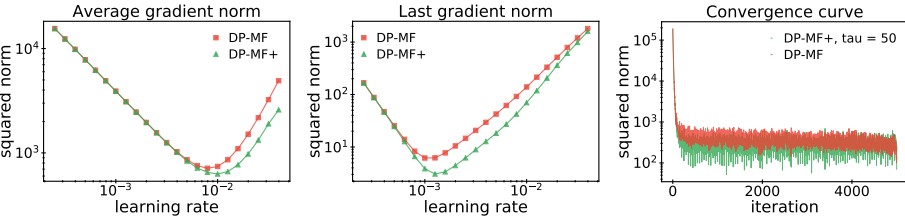

(a) Average gradient norm for varying learning rates.
(b) Last gradient norm for varying learning rates.
(c) Gradient norm over time for $\gamma = 10^{-2}$.

Figure 2: Comparison of the average and last gradient norms for DP-MF and DP-MF$^+$ on a random non-strongly convex quadratic function with $L = 10$.

where $\mathbf{\Lambda}_\tau = \left[ \boldsymbol{\lambda}_1^\top, \ldots, \boldsymbol{\lambda}_T^\top \right]^\top \in \mathbb{R}^{T \times T}$, and we set the rows $\boldsymbol{\lambda}_t$ appropriately to select corresponding row differences of $\mathbf{B}$ with either coefficient $1$ or $1/\sqrt{\tau}$ depending on the index $t$. We give a precise definition of $\mathbf{\Lambda}_\tau$ and an explicit example when $T = 12, \tau = 3$ in Appendix B.

Recall that [10] minimize the Frobenius norm objective (6) based on their derived convergence bounds in (5). Since our derived convergence bounds are strictly tighter, we propose using Eq. (15) as the new objective function in (6). Intuitively, since $\|\mathbf{\Lambda}_\tau \mathbf{B}\|_F^2$ is a better proxy for learning performance than $\|\mathbf{B}\|_F^2$, minimizing this quantity in the offline factorization problem should lead to ERM methods with better privacy-utility trade-offs.

We can solve our new offline matrix factorization problem in a straightforward manner. We can show that for $\mathbf{A} = \mathbf{S}$, we can solve this modified problem by first computing the solution $\tilde{\mathbf{B}}, \tilde{\mathbf{C}}$ using $\mathrm{OPT}_F(\mathbf{\Lambda}_\tau \mathbf{A})$. The solution to our modified objective is then $\mathbf{C} = \tilde{\mathbf{C}}, \mathbf{B} = \mathbf{A}\mathbf{C}^{-1}$. This implies we can use existing open-source solvers designed for (6) [51, 30, 10].

## 6 Experiments

In this section, we evaluate the ERM performance of MF-DP-FTRL under different offline factorization objectives. We focus on the Frobenius norm objective (6), which we refer to as DP-MF [10, 7], and our modified objective (15), which we refer to as DP-MF$^+$.

### 6.1 Validating Theoretical Results

We first validate our theoretical results above by comparing the convergence of DP-MF and DP-MF$^+$ on a *random quadratic* function that satisfies the assumptions of Theorem 4.7. Notably, we ensure the quadratic is not strongly convex. We treat $\tau$ in (15) as a hyperparameter and tune it over a fixed grid. For complete details, please refer to Appendix H. We present the results in Fig. 2.

In Fig. 2(a) we plot $\frac{1}{T} \sum_{t=0}^{T} \|\nabla f(\mathbf{x}_t)\|^2$, as this quantity is proportional to the LHS of Theorem 4.7. For all learning rates, DP-MF$^+$ either matches or outperforms DP-MF. Moreover, the advantage of DP-MF$^+$ increases as the learning rate increases. This corresponds to our theory in Theorem 4.7. Indeed, the larger the stepsize $\gamma$, the smaller the optimal $\tau$ (as $\tau = \Theta\left(1/\gamma L\right)$), and the more often restarts are used in the analysis of Theorem 4.7.

Fig. 2(b) further depicts the last-iterate behaviours of DP-MF and DP-MF$^+$, which is often more practically relevant. Interestingly, the last iterate behaviour is improved even in the cases where the average behaviour does not improve. Finally, in Fig. 2(c) we pick $\gamma = 10^{-2}, \tau = 50$ as the parameters for which both the average and the last-iterate behaviours are improved and plot the convergence curve over iterations. DP-MF$^+$ has regular oscillating behaviour, allowing it to achieve a good final-iterate performance. The period of these oscillations is exactly equal to $\tau$.

### 6.2 Practical DP Training Experiments

We now compare DP-MF, DP-MF$^+$, and DP-SGD with privacy amplification [1] on the MNIST, CIFAR-10, and Stack Overflow datasets. We omit from comparison DP-FTRL [20] and DP-Fourier

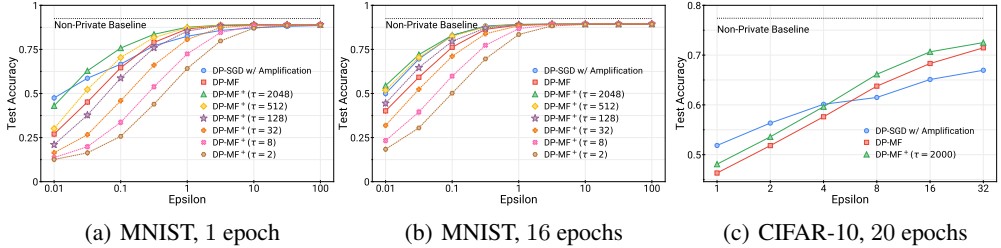

| (a) MNIST, 1 epoch | (b) MNIST, 16 epochs | (c) CIFAR-10, 20 epochs |

Figure 3: Test set accuracy of various mechanisms on the MNIST and CIFAR-10 datasets.

[7] as these methods are strictly dominated by DP-MF. Unlike our theoretical analysis, we include clipping to derive formal $(\varepsilon, \delta)$ privacy guarantees. To facilitate a fair comparison, we set $\delta = 10^{-6}$ in all the settings, and compare against varying $\varepsilon$. We give complete experimental details in Appendix H

**MNIST, logistic regression.** We train for $T = 2048$ iterations and either 1 or 16 epochs depending on the batch size, corresponding to a batch size of 29 and 469 respectively.[2] We fix the clipping threshold at $1.0$ and the learning rate at $0.5$. We vary $\tau$ in (15) over $\{2, 8, 32, 128, 512, 2048\}$. The results are in Figs. 3(a) and 3(b). DP-MF$^+$ improves monotonically with $\tau$, performing best when $\tau = 2048 = T$. For such $\tau$, DP-MF$^+$ consistently out-performs DP-MF across all settings. Recall from (15) that this corresponds to the offline objective $\|\mathbf{\Lambda}_T \mathbf{B}\|_F^2$ where $\lambda_{ii} = 1/\sqrt{T}$ for all $i < T$ and $\lambda_{TT} = 1$. This objective strongly penalizes errors on the final iterate, which is the model used to compute test accuracy.

We also see that DP-MF$^+$ expands the number of settings in which we can beat DP-SGD. DP-MF only outperforms DP-SGD for sufficiently large $\varepsilon$ ($\varepsilon \geq 0.31$ for 1 epoch and $\varepsilon \geq 31$ for 16 epochs). By contrast, DP-MF$^+$ outperforms DP-SGD in every setting except when $\varepsilon = 0.01$ and 1 epoch. None of the mechanisms reached the accuracy levels obtained by the non-private baseline, even at $\varepsilon = 100$. We suspect this is due to the fact that we are using a fixed but aggressive clipping threshold of $1.0$ across all experiments, which helps in the moderate privacy regime but hurts in very low privacy regime. Even though DP-MF$^+$ does not use privacy amplification, it outperforms DP-SGD, which uses privacy amplification. This is due to the efficient noise anti-correlations in DP-MF$^+$. If amplification were not possible, performance of DP-SGD would degrade even further.

**CIFAR-10, CNN.** We follow the experimental setup from [7]. Specifically, we train all mechanisms for 20 epochs and $T = 2000$ iterations, which corresponds to a batch size of $500$.[3] We tune the learning rate over a fixed grid. We fix $\tau = T = 2000$ in DP-MF$^+$ as we found that worked best in the MNIST experiments. The results are given in Fig. 3(c). We see that DP-MF$^+ (\tau = 2000)$ offers a consistent improvement over DP-MF across all choices of $\varepsilon$ considered. Both DP-MF and DP-MF$^+$ beat DP-SGD for $\varepsilon > 4$. This observation is consistent with prior work on DP-FTRL and DP-MF, where DP-SGD performs relatively better with smaller $\varepsilon$ while DP-MF performs better with larger $\varepsilon$.

**Stack Overflow, LSTM.** In Appendix H, we compare DP-MF and DP-MF$^+$ on a federated learning task with *user-level* differential privacy. We do not compare DP-SGD on this task, as amplification techniques such as shuffling and subsampling are not possible in practical federated learning settings [20]. In this task, we train an LSTM network to do next-word prediction on the Stack Overflow dataset. To be consistent with the prior work [10] and to test if our proposed factorizations are compatible with the other types of workloads $\mathbf{A}$ from Eq. (2), we use momentum and learning rate decay. Our results are given in Table 2. We see that two methods perform comparably, verifying competitiveness of our method. Note that this task uses federated averaging [32] instead of gradient

---

[2]In practice, one often trains small-scale models for many epochs, perhaps even using full-batch gradients, to improve the privacy/utility trade-off (at the cost of increased computation). We are interested in the *relative* performance for a fixed computation budget, so we train for a small number of epochs.

[3]While Choquette-Choo et al. [7] use momentum and learning rate decay, we omit the use of such techniques as they are orthogonal to our theoretical results.

descent. Developing offline factorization objectives specifically for federated learning remains an open problem.

# 7 Conclusion

In this work, we developed analytic techniques to study the convergence of gradient descent under linearly correlated noise that is motivated from a class of DP mechanisms. We derived tighter bounds than currently exist in the literature, and we use our novel theoretical understanding to design privacy mechanisms with improved convergence. Perhaps more importantly, our work highlights the wealth of stochastic optimization questions arising from recent advances in differentially private model training. As such, we distill and formalize various optimization problems arising from recent work on matrix mechanisms for DP. Our work raises a host of questions and open problems, including extending our analysis to include things such as clipping, shuffling, and momentum. Another key extension is to derive last-iterate convergence rates rather than average-iterate convergence rates, as in some settings it is only the final "released" model that needs formal privacy guarantees. Given the improved generalization properties of Anti-PGD [37], one could also investigate how to design more general linearly correlated noise mechanisms which improve both privacy and generalization.

# 8 Acknowledgments

The authors would like to thank Francesco D'Angelo, Nina Mainusch and Linara Adylova for their comments on the manuscript. The authors would also like to thank the reviewers for their helpful suggestions in improving the clarity of the writing.

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
