# A Additional Examples

## A.1 Why the Frobenius Norm is not Predictive

In this section we give an explicit example of a matrix $\mathbf{B}$ for which the Frobenius norm $\|\mathbf{B}\|_F$ does not give a good estimation of the optimization behavior of (7).

**Example A.1** (Chess-PGD). We consider the special case of algorithm (7) whose noise correlation matrix $\mathbf{B}$ whose lower triangle has a chess board-like structure given by

$$\mathbf{B}_{\text{chess}} = \sqrt{2} \begin{pmatrix} 1 & 0 & 0 & \ldots & 0 \\ 0 & 1 & 0 & \ldots & 0 \\ 1 & 0 & 1 & \ldots & 0 \\ \ldots & & & & \\ 0 & 1 & 0 & \ldots & 1 \end{pmatrix}$$

We refer to this algorithm (whose perturbed noise structure is given by $\mathbf{B}_{\text{chess}}$) as Chess-PGD. Note that $\text{sens}(\mathbf{C}_{\text{chess}}) \|\mathbf{B}_{\text{chess}}\|_F = \text{sens}(\mathbf{C}_{\mathbf{S}}) \|\mathbf{S}\|_F$. Despite this, PGD (for which $\mathbf{B} = \mathbf{S}$) converges strictly faster than Chess-PGD in Fig. 4.

By contrast, our Theorem 4.7 is better able to capture the behaviour of Chess-PGD. Suppose that $\tau \leq T/4$. Given a row $\mathbf{b}_t$ of $\mathbf{B}_{\text{chess}}$, for any $t' < t$ we have

$$\frac{t - t'}{2} \leq \|\mathbf{b}_t - \mathbf{b}_{t'}\|^2 \leq t.$$

Therefore, at least $T/4$ of the summands in the noise term of Theorem 4.7 are on the order of $\Theta(T)$. Plugging in this estimate into the convergence rate, we find that Chess-PGD produces iterates that satisfy the convergence rate

$$\frac{1}{T+1} \sum_{t=0}^{T} \mathbb{E}\left[f(\mathbf{x}_t) - f^*\right] = \tilde{\mathcal{O}}\left(\frac{\|\mathbf{x}_0 - \mathbf{x}^\star\|^2}{\gamma T} + LT\gamma^2\sigma^2\right). \tag{16}$$

Indeed, as we show below (and plot in Figure 4), Chess-PGD linearly diverges with $T$ as predicted.

## A.2 Experimental Comparison of PGD with Chess-PGD

In this section we illustrate that Chess-PGD diverges while PGD converges for the same quadratic functions as in Section 6. We set the stepsize constant, $\gamma = 0.02$. We plot $\|\nabla f(\mathbf{x}_t)\|^2$ at each iteration $t$. We see that, as predicted by (16), Chess-PGD diverges with linear rate in $T$, while PGD converges to a constant noise level.

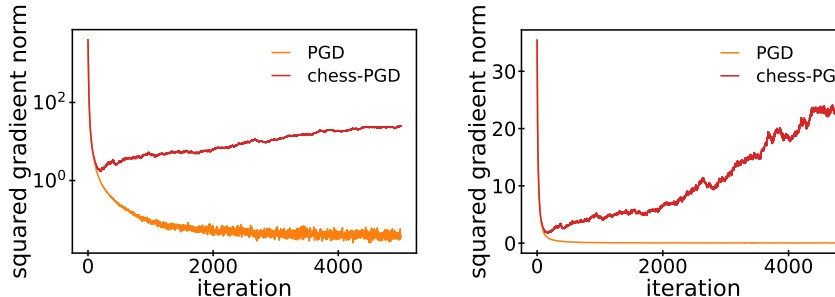

Figure 4: Comparison of PGD and Chess-PGD under the fixed stepsize, $\gamma = 0.02$. Y axis in the log scale on the left, and in the normal scale on the right.

# B Factorization Matrices

As discussed in Section 2, Denisov et al. [10] propose finding useful factorizations for DP training by solving the problem

$$\min_{\mathbf{B},\mathbf{C}} \|\mathbf{B}\|_F^2 \quad \text{such that} \quad \mathbf{BC} = \mathbf{A}, \; \text{sens}(\mathbf{C}) = 1. \tag{17}$$

As we discuss in Section 5, based on our convergence rates in Section 4, we propose the following modified objective:

$$\min_{\mathbf{B},\mathbf{C}} \|\Lambda_\tau \mathbf{B}\|_F^2 \quad \text{such that} \ \ \mathbf{BC} = \mathbf{A}, \ \text{sens}(\mathbf{C}) = 1. \tag{18}$$

The matrix $\Lambda_\tau = [\lambda_{tj}]_{t,j=1,\dots,T}$ is defined as follows:

$$\lambda_{tj} = \begin{cases} \frac{1}{\sqrt{\tau}} & j = t, & t \neq 0 \bmod \tau \\ -\frac{1}{\sqrt{\tau}} & j = \lfloor \frac{t}{\tau} \rfloor \tau, & t \neq 0 \bmod \tau, t > \tau \\ 1 & j = t, & t = 0 \bmod \tau \\ -1 & j = t - \tau, & t = 0 \bmod \tau, t > \tau \end{cases}$$

For all the other indices, $\lambda_{tj} = 0$. In Figure 5 we give an example of such a matrix for $T = 12$ and $\tau = 3$.

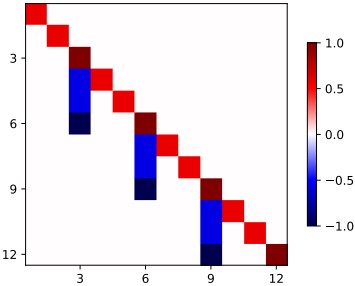

Figure 5: Elements of $\Lambda_\tau$ for $T = 12$, and $\tau = 3$.

To illustrate how the parameter $\tau$ affects the solution to the objective problem, we plot numerically computed approximate minimizers to (17) and (18) in Figure 6(a) and Figure 6(b), respectively. Specifically, we plot the matrix $\mathbf{B}$, and let $\mathbf{B}_{\text{MF}}$ denote the solution to (17) and $\mathbf{B}_{\text{MF+}}$ denote the solution to (18). We can clearly see that for the latter, the parameter $\tau$ enforces a block-like structure such that the bands of correlation are at regular intervals of length $\tau$.

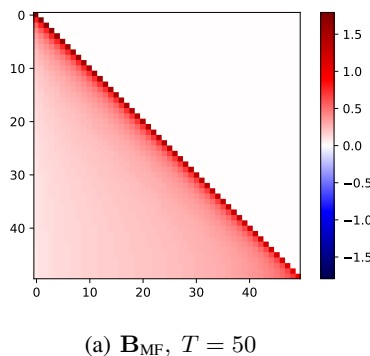

(a) $\mathbf{B}_{\text{MF}}$, $T = 50$

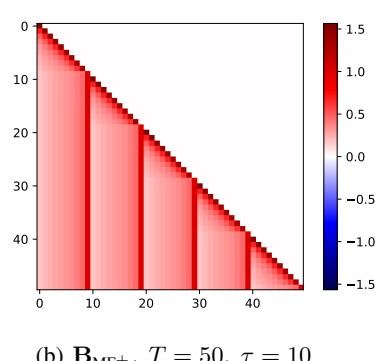

(b) $\mathbf{B}_{\text{MF+}}$, $T = 50$, $\tau = 10$

## C  Proofs of Main Results

We analyse the algorithm with general $\mathbf{B}$ that has the following iterates:

$$\mathbf{x}_{t+1} = \mathbf{x}_t - \eta(\nabla f(\mathbf{x}_t) + (\mathbf{b}_{t+1} - \mathbf{b}_t)^\top \mathbf{Z}) \quad t \geq 1 \tag{19}$$

where $\mathbf{b}_0 = 0$. We define $\mathbf{v}_t = (\mathbf{b}_{t+1} - \mathbf{b}_t)^\top \mathbf{Z}$ for $t \geq 0$, so that

$$\mathbf{x}_{t+1} = \mathbf{x}_t - \gamma \nabla f(\mathbf{x}_t) - \gamma \mathbf{v}_t$$

For the analysis, we define a virtual sequence with restarts (14), where we do restarts every $\tau$ iterations. Formally, we define virtual iterates $\{\tilde{\mathbf{x}}_t\}_{t=0}^T$ as follows:

$$\tilde{\mathbf{x}}_{t+1} = \tilde{\mathbf{x}}_t - \gamma\nabla f(\mathbf{x}_t) \qquad \text{if } t+1 \neq 0 \bmod \tau$$
$$\tilde{\mathbf{x}}_{t+1} = \mathbf{x}_{t+1} \qquad \text{if } t+1 = 0 \bmod \tau \text{ (restart iterations)}$$

This means that $\tilde{\mathbf{x}}_{k\tau} = \mathbf{x}_{k\tau}$, for any nonnegative integer $k$.

**Useful facts about this sequence.**

- The closest restart iteration to $t$ is equal to $\lfloor\frac{t}{\tau}\rfloor\tau$.
- For $t < \tau$ we have

$$\tilde{\mathbf{x}}_t - \mathbf{x}_t = \gamma\mathbf{b}_t^\top\mathbf{Z}$$

- For restart iterations $t + 1 = \tau$,

$$\tilde{\mathbf{x}}_{t+1} - \mathbf{x}_{t+1} = 0$$

- For the next iteration just after restart $t + 1 = \tau + 1$

$$\tilde{\mathbf{x}}_{\tau+1} - \mathbf{x}_{\tau+1} = (\tilde{\mathbf{x}}_\tau - \gamma\nabla f(\mathbf{x}_\tau)) - (\mathbf{x}_\tau - \gamma\nabla f(\mathbf{x}_\tau) - \gamma\mathbf{v}_\tau) = \gamma\mathbf{v}_\tau = \gamma(\mathbf{b}_{\tau+1} - \mathbf{b}_\tau)^\top\mathbf{Z}$$

- Thus, for arbitrary $t$,

$$\tilde{\mathbf{x}}_t - \mathbf{x}_t = \gamma(\mathbf{b}_t - \mathbf{b}_{\lfloor\frac{t}{\tau}\rfloor\tau})^\top\mathbf{Z} \tag{20}$$

  (and if $t = 0 \bmod \tau$, then the term cancels and we get $\tilde{\mathbf{x}}_t - \mathbf{x}_t = 0$), we assume that $\mathbf{b}_0 = \mathbf{0}$.
- We can re-write the restart iterations for $t + 1 = 0 \bmod \tau$

$$\tilde{\mathbf{x}}_{t+1} = \mathbf{x}_{t+1} = \mathbf{x}_t - \gamma\nabla f(\mathbf{x}_t) - \gamma(\mathbf{b}_{t+1} - \mathbf{b}_t)^\top\mathbf{Z}$$
$$= \tilde{\mathbf{x}}_t - \gamma\nabla f(\mathbf{x}_t) - \gamma(\mathbf{b}_t - \mathbf{b}_{\lfloor\frac{t}{\tau}\rfloor\tau})^\top\mathbf{Z} - \gamma(\mathbf{b}_{t+1} - \mathbf{b}_t)^\top\mathbf{Z}$$
$$= \tilde{\mathbf{x}}_t - \gamma\nabla f(\mathbf{x}_t) - \gamma(\mathbf{b}_{t+1} - \mathbf{b}_{\lfloor\frac{t}{\tau}\rfloor\tau})^\top\mathbf{Z}$$

Equivalently, for $t + 1 = 0 \bmod \tau$,

$$\tilde{\mathbf{x}}_{t+1} = \tilde{\mathbf{x}}_t - \gamma\nabla f(\mathbf{x}_t) - \gamma(\mathbf{b}_{t+1} - \mathbf{b}_{t+1-\tau})^\top\mathbf{Z}. \tag{21}$$

## C.1 Assumptions and Useful Inequalities

This section contains assumptions and inequalities that will be used throughout the proof. First, recall that in Assumption 4.2, we assume that $f$ is differentiable and $L$-smooth, so that

$$\forall\mathbf{x}, \mathbf{y} \in \mathbb{R}^d, \quad \|\nabla f(\mathbf{x}) - \nabla f(\mathbf{y})\| \leq L\|\mathbf{x} - \mathbf{y}\|. \tag{22}$$

In some settings, we will also assume convexity, so that

$$\forall\mathbf{x}, \mathbf{y} \in \mathbb{R}^d, \quad f(\mathbf{x}) - f(\mathbf{y}) \leq \langle\nabla f(\mathbf{x}), \mathbf{x} - \mathbf{y}\rangle. \tag{23}$$

We will also make use of the following facts about the geometry of vectors in $\mathbb{R}^d$.

**Lemma C.1.** *For any finite set of vectors $\{\mathbf{a}_i\}_{i=1}^n \subset \mathbb{R}^d$,*

$$\left\|\sum_{i=1}^n \mathbf{a}_i\right\|^2 \leq n\sum_{i=1}^n \|\mathbf{a}_i\|^2. \tag{24}$$

**Lemma C.2.** *For any two vectors $\mathbf{a}, \mathbf{b} \in \mathbb{R}^d$ and for all $\alpha > 0$,*

$$2\langle\mathbf{a}, \mathbf{b}\rangle \leq \alpha\|\mathbf{a}\|^2 + \alpha^{-1}\|\mathbf{b}\|^2. \tag{25}$$

## C.2 Proof for Non-convex Functions

**Iterations without restarts.** If $t$ is such that $t \neq -1 \bmod \tau$, where $k$ is some integer number, then between iteration $t$ and $t+1$ no restart of virtual sequence happens and thus $\tilde{\mathbf{x}}_{t+1} = \tilde{\mathbf{x}}_t - \gamma \nabla f(\mathbf{x}_t)$. We follow closely standard perturbed iterate analysis [28, 43]. By $L$-smoothness of $f$

$$f(\tilde{\mathbf{x}}_{t+1}) \leq f(\tilde{\mathbf{x}}_t) - \gamma \langle \nabla f(\tilde{\mathbf{x}}_t), \nabla f(\mathbf{x}_t) \rangle + \frac{L\gamma^2}{2} \|\nabla f(\mathbf{x}_t)\|^2$$

$$\leq f(\tilde{\mathbf{x}}_t) - \frac{\gamma}{2} \|\nabla f(\tilde{\mathbf{x}}_t)\|^2 - \frac{\gamma}{2} \|\nabla f(\mathbf{x}_t)\|^2 + \frac{\gamma L^2}{2} \|\mathbf{x}_t - \tilde{\mathbf{x}}_t\|^2$$

$$\overset{(20)}{\leq} f(\tilde{\mathbf{x}}_t) - \frac{\gamma}{2} \|\nabla f(\tilde{\mathbf{x}}_t)\|^2 - \frac{\gamma}{2} \|\nabla f(\mathbf{x}_t)\|^2 + \frac{\gamma^3 L^2}{2} \left\| (\mathbf{b}_t - \mathbf{b}_{\lfloor \frac{t}{\tau} \rfloor \tau})^\top \mathbf{Z} \right\|^2 \quad (26)$$

where on the second line we used that $-2\langle \mathbf{a}, \mathbf{b} \rangle = - \|\mathbf{a}\|^2 - \|\mathbf{b}\|^2 + \|\mathbf{a} - \mathbf{b}\|^2$ for any $\mathbf{a}, \mathbf{b} \in \mathbb{R}^d$.

**Iterations with restarts.** Restart happens between iteration $t$ and $t+1$ if $t = -1 \bmod \tau$. In this case, the analysis is more involved. By $L$-smoothness and using update rule (21)

$$f(\tilde{\mathbf{x}}_{t+1}) \leq f(\tilde{\mathbf{x}}_t) - \gamma \langle \nabla f(\tilde{\mathbf{x}}_t), \nabla f(\mathbf{x}_t) + (\mathbf{b}_{t+1} - \mathbf{b}_{t+1-\tau})^\top \mathbf{Z} \rangle \quad (27)$$

$$+ \frac{L}{2} \gamma^2 \left\| \nabla f(\mathbf{x}_t) + (\mathbf{b}_{t+1} - \mathbf{b}_{t+1-\tau})^\top \mathbf{Z} \right\|^2 \quad (28)$$

$$\overset{(24)}{\leq} f(\tilde{\mathbf{x}}_t) - \underbrace{\gamma \langle \nabla f(\tilde{\mathbf{x}}_t), \nabla f(\mathbf{x}_t) \rangle}_{:=T_1} - \underbrace{\gamma \langle \nabla f(\tilde{\mathbf{x}}_t), (\mathbf{b}_{t+1} - \mathbf{b}_{t+1-\tau})^\top \mathbf{Z} \rangle}_{:=T_2} \quad (29)$$

$$+ L\gamma^2 \|\nabla f(\mathbf{x}_t)\|^2 + L\gamma^2 \left\| (\mathbf{b}_{t+1} - \mathbf{b}_{t+1-\tau})^\top \mathbf{Z} \right\|^2 \quad (30)$$

We estimate separately the second and the third terms

$$T_1 = -\frac{\gamma}{2} \|\nabla f(\tilde{\mathbf{x}}_t)\|^2 - \frac{\gamma}{2} \|\nabla f(\mathbf{x}_t)\|^2 + \frac{\gamma}{2} \|\nabla f(\tilde{\mathbf{x}}_t) - \nabla f(\mathbf{x}_t)\|^2$$

$$\overset{(22)}{\leq} -\frac{\gamma}{2} \|\nabla f(\tilde{\mathbf{x}}_t)\|^2 - \frac{\gamma}{2} \|\nabla f(\mathbf{x}_t)\|^2 + \frac{\gamma L^2}{2} \|\tilde{\mathbf{x}}_t - \mathbf{x}_t\|^2$$

$$\overset{(20)}{\leq} -\frac{\gamma}{2} \|\nabla f(\tilde{\mathbf{x}}_t)\|^2 - \frac{\gamma}{2} \|\nabla f(\mathbf{x}_t)\|^2 + \frac{\gamma^3 L^2}{2} \left\| (\mathbf{b}_t - \mathbf{b}_{\lfloor \frac{t}{\tau} \rfloor \tau})^\top \mathbf{Z} \right\|^2$$

The third term,

$$T_2 = -\langle \nabla f(\tilde{\mathbf{x}}_t), \gamma (\mathbf{b}_{t+1} - \mathbf{b}_{t+1-\tau})^\top \mathbf{Z} \rangle$$

$$\overset{(25), \alpha = \frac{1}{8L}}{\leq} \frac{1}{16L} \|\nabla f(\tilde{\mathbf{x}}_t)\|^2 + 4L\gamma^2 \left\| (\mathbf{b}_{t+1} - \mathbf{b}_{t+1-\tau})^\top \mathbf{Z} \right\|^2$$

It is left to deal with the norm of the gradient $\frac{1}{16L}\|\nabla f(\tilde{\mathbf{x}}_t)\|^2$. Using that $\tau = \frac{1}{L\gamma}$, and thus $\frac{1}{16L\tau} = \frac{\gamma}{16}$ we have

$$\frac{1}{16L}\|\nabla f(\tilde{\mathbf{x}}_t)\|^2 = \frac{\gamma}{16}\sum_{i=0}^{\tau-1}\|\nabla f(\tilde{\mathbf{x}}_t)\|^2$$

$$\overset{(24),(22)}{\leq} \frac{\gamma}{8}\sum_{i=0}^{\tau-1}L^2\|\tilde{\mathbf{x}}_t - \tilde{\mathbf{x}}_{t-i}\|^2 + \frac{\gamma}{8}\sum_{i=0}^{\tau-1}\|\nabla f(\tilde{\mathbf{x}}_{t-i})\|^2$$

$$\overset{(26)}{\leq} \frac{\gamma}{8}\sum_{i=1}^{\tau-1}\gamma^2 L^2\left\|\sum_{j=t-i}^{t-1}\nabla f(\mathbf{x}_j)\right\|^2 + \frac{\gamma}{8}\sum_{i=0}^{\tau-1}\|\nabla f(\tilde{\mathbf{x}}_{t-i})\|^2$$

$$\overset{(24)}{\leq} \frac{\gamma^3 L^2}{8}\sum_{i=1}^{\tau-1}\tau\sum_{j=t-i}^{t-1}\|\nabla f(\mathbf{x}_j)\|^2 + \frac{\gamma}{8}\sum_{i=0}^{\tau-1}\|\nabla f(\tilde{\mathbf{x}}_{t-i})\|^2$$

$$\leq \frac{\gamma^3 L^2 \tau^2}{8}\sum_{i=1}^{\tau-1}\|\nabla f(\mathbf{x}_{t-i})\|^2 + \frac{\gamma}{8}\sum_{i=0}^{\tau-1}\|\nabla f(\tilde{\mathbf{x}}_{t-i})\|^2$$

$$\overset{\tau=\frac{1}{\gamma L}}{\leq} \frac{\gamma}{8}\sum_{i=1}^{\tau-1}\|\nabla f(\mathbf{x}_{t-i})\|^2 + \frac{\gamma}{8}\sum_{i=0}^{\tau-1}\|\nabla f(\tilde{\mathbf{x}}_{t-i})\|^2$$

Putting back our calculations of $T_1$ and $T_2$ into (30), and setting $\gamma \leq \frac{1}{4L}$ in order to estimate that $L\gamma^2\|\nabla f(\mathbf{x}_t)\|^2 \leq \frac{\gamma}{4}\|\nabla f(\mathbf{x}_t)\|^2$

$$f(\tilde{\mathbf{x}}_{t+1}) \leq f(\tilde{\mathbf{x}}_t) - \frac{\gamma}{2}\|\nabla f(\tilde{\mathbf{x}}_t)\|^2 - \frac{\gamma}{4}\|\nabla f(\mathbf{x}_t)\|^2 + \frac{\gamma^3 L^2}{2}\left\|(\mathbf{b}_t - \mathbf{b}_{\lfloor\frac{t}{\tau}\rfloor\tau})^\top\mathbf{Z}\right\|^2$$

$$+ 5L\gamma^2\left\|(\mathbf{b}_{t+1} - \mathbf{b}_{t+1-\tau})^\top\mathbf{Z}\right\|^2 + \frac{\gamma}{8}\sum_{i=1}^{\tau-1}\|\nabla f(\mathbf{x}_{t-i})\|^2 + \frac{\gamma}{8}\sum_{i=0}^{\tau-1}\|\nabla f(\tilde{\mathbf{x}}_{t-i})\|^2$$
(31)

**Combining iterations with and without restarts.** It is left to average equations (26) and (31) over all iterations $0 \leq t \leq T$. We denote $\mathcal{T}_1$ is the set of indices without restarts, and $\mathcal{T}_2$ are restarts indices.

$$\sum_{t\in\mathcal{T}_1}\frac{\gamma}{8}\left(\|\nabla f(\tilde{\mathbf{x}}_t)\|^2 + \|\nabla f(\mathbf{x}_t)\|^2\right) + \sum_{t\in\mathcal{T}_2}\frac{\gamma}{8}\left(\|\nabla f(\tilde{\mathbf{x}}_t)\|^2 + \|\nabla f(\mathbf{x}_t)\|^2\right)$$

$$\leq (f(\mathbf{x}_0) - f^\star) + \frac{\gamma^3 L^2}{2}\sum_{t=1}^{T}\left\|(\mathbf{b}_t - \mathbf{b}_{\lfloor\frac{t}{\tau}\rfloor\tau})^\top\mathbf{Z}\right\|^2 + 5L\gamma^2\sum_{t\in\mathcal{T}_1}\left\|(\mathbf{b}_{t+1} - \mathbf{b}_{t+1-\tau})^\top\mathbf{Z}\right\|^2$$

Dividing by $\frac{\gamma(T+1)}{8}$, we get

$$\frac{1}{T+1}\sum_{t=0}^{T}\mathbb{E}\|\nabla f(\mathbf{x}_t)\|^2 \leq \frac{8(f(\mathbf{x}_0) - f^\star)}{\gamma(T+1)} + \frac{4\gamma^2 L^2}{T+1}\sum_{t=1}^{T}\mathbb{E}\left\|(\mathbf{b}_t - \mathbf{b}_{\lfloor\frac{t}{\tau}\rfloor\tau})^\top\mathbf{Z}\right\|^2$$

$$+ \frac{40L\gamma}{T+1}\sum_{k=1}^{\lfloor\frac{T}{\tau}\rfloor}\mathbb{E}\left\|(\mathbf{b}_{k\tau} - \mathbf{b}_{(k-1)\tau})^\top\mathbf{Z}\right\|^2$$

which completes the proof.

### C.3  Proof for Convex Functions

Our proof for convex functions follows the same pattern as for non-convex: we consider separately iterations with and without restarts of the virtual sequence (14). However, summing up these two cases is the most involved part of the proof in the convex case, and it is different from the non-convex case.

We will use the following fact in our proof.

**Lemma C.3.** *If function $f$ is convex (23), $L$-smooth (22), and has a finite minimizer $x^*$, then*

$$\|\nabla f(\mathbf{x})\|^2 \leq 2L\left(f(\mathbf{x}) - f^\star\right). \tag{32}$$

**Iterations without restarts.** Using (14), i.e. that $\tilde{\mathbf{x}}_{t+1} = \tilde{\mathbf{x}}_t - \gamma \nabla f(\mathbf{x}_t)$, for some point $\mathbf{x}^\star$ that satisfies $\nabla f(\mathbf{x}^\star) = 0$,

$$\|\tilde{\mathbf{x}}_{t+1} - \mathbf{x}^\star\|^2 = \|\tilde{\mathbf{x}}_t - \mathbf{x}^\star\|^2 - 2\gamma\langle\nabla f(\mathbf{x}_t), \mathbf{x}_t - \mathbf{x}^\star\rangle + \gamma^2\|\nabla f(\mathbf{x}_t)\|^2 + 2\gamma\langle\nabla f(\mathbf{x}_t), \mathbf{x}_t - \tilde{\mathbf{x}}_t\rangle$$

$$\overset{(32),(23)}{\leq} \|\tilde{\mathbf{x}}_t - \mathbf{x}^\star\|^2 - 2\gamma(1 - L\gamma)\left(f(\mathbf{x}_t) - f^\star\right) + 2\gamma\langle\nabla f(\mathbf{x}_t), \mathbf{x}_t - \tilde{\mathbf{x}}_t\rangle$$

We estimate the last term separately

$$2\langle\nabla f(\mathbf{x}_t), \mathbf{x}_t - \tilde{\mathbf{x}}_t\rangle \overset{(25),\alpha=2L}{\leq} \frac{1}{2L}\|\nabla f(\mathbf{x}_t)\|^2 + 2L\|\mathbf{x}_t - \tilde{\mathbf{x}}_t\|^2 \overset{(32)}{\leq} \left(f(\mathbf{x}_t) - f^\star\right) + 2L\|\mathbf{x}_t - \tilde{\mathbf{x}}_t\|^2$$

Thus,

$$\|\tilde{\mathbf{x}}_{t+1} - \mathbf{x}^\star\|^2 \leq \|\tilde{\mathbf{x}}_t - \mathbf{x}^\star\|^2 - \gamma(1 - 2L\gamma)(f(\mathbf{x}_t) - f^\star) + 2L\gamma\|\mathbf{x}_t - \tilde{\mathbf{x}}_t\|^2$$

$$\overset{\gamma < \frac{1}{4L},(20)}{\leq} \|\tilde{\mathbf{x}}_t - \mathbf{x}^\star\|^2 - \frac{\gamma}{2}(f(\mathbf{x}_t) - f^\star) + 2L\gamma^3 \left\|(\mathbf{b}_t - \mathbf{b}_{\lfloor\frac{t}{\tau}\rfloor\tau})^\top \mathbf{Z}\right\|^2 \tag{33}$$

**For the iterations with restarts.** This means that $t + 1 = k\tau$. Using (21),

$$\|\tilde{\mathbf{x}}_{t+1} - \mathbf{x}^\star\|^2 = \left\|\tilde{\mathbf{x}}_t - \mathbf{x}^\star - \gamma\nabla f(\mathbf{x}_t) - \gamma(\mathbf{b}_{t+1} - \mathbf{b}_{t+1-\tau})^\top\mathbf{Z}\right\|^2$$

$$= \|\tilde{\mathbf{x}}_t - \mathbf{x}^\star\|^2 - 2\gamma\langle\nabla f(\mathbf{x}_t), \tilde{\mathbf{x}}_t - \mathbf{x}^\star\rangle - 2\gamma\langle(\mathbf{b}_{t+1} - \mathbf{b}_{t+1-\tau})^\top\mathbf{Z}, \tilde{\mathbf{x}}_t - \mathbf{x}^\star\rangle$$

$$+ \gamma^2\left\|\nabla f(\mathbf{x}_t) + (\mathbf{b}_{t+1} - \mathbf{b}_{t+1-\tau})^\top\mathbf{Z}\right\|^2$$

We estimate the second term same as in the case without restarts:

$$-2\gamma\langle\nabla f(\mathbf{x}_t), \tilde{\mathbf{x}}_t - \mathbf{x}^\star\rangle = -2\gamma\langle\nabla f(\mathbf{x}_t), \mathbf{x}_t - \mathbf{x}^\star\rangle - 2\gamma\langle\nabla f(\mathbf{x}_t), \tilde{\mathbf{x}}_t - \mathbf{x}_t\rangle$$

$$\leq -\gamma(f(\mathbf{x}_t) - f^\star) + 2L\gamma\|\mathbf{x}_t - \tilde{\mathbf{x}}_t\|^2$$

For the last term,

$$\gamma^2\left\|\nabla f(\mathbf{x}_t) + (\mathbf{b}_{t+1} - \mathbf{b}_{t+1-\tau})^\top\mathbf{Z}\right\|^2 \overset{(24)}{\leq} 2\gamma^2\|\nabla f(\mathbf{x}_t)\|^2 + 2\gamma^2\left\|(\mathbf{b}_{t+1} - \mathbf{b}_{t+1-\tau})^\top\mathbf{Z}\right\|^2$$

$$\overset{(32)}{\leq} 4L\gamma^2(f(\mathbf{x}_t) - f^\star) + 2\gamma^2\left\|(\mathbf{b}_{t+1} - \mathbf{b}_{t+1-\tau})^\top\mathbf{Z}\right\|^2$$

Thus with $\gamma \leq \frac{1}{8L}$,

$$\frac{\gamma}{2}(f(\mathbf{x}_t) - f^\star) \leq \|\tilde{\mathbf{x}}_t - \mathbf{x}^\star\|^2 - \|\tilde{\mathbf{x}}_{t+1} - \mathbf{x}^\star\|^2 + 2L\gamma^3\left\|(\mathbf{b}_t - \mathbf{b}_{\lfloor\frac{t}{\tau}\rfloor\tau})^\top\mathbf{Z}\right\|^2$$

$$+ 2\gamma^2\left\|(\mathbf{b}_{t+1} - \mathbf{b}_{t+1-\tau})^\top\mathbf{Z}\right\|^2 - 2\gamma\langle(\mathbf{b}_{t+1} - \mathbf{b}_{t+1-\tau})^\top\mathbf{Z}, \tilde{\mathbf{x}}_t - \mathbf{x}^\star\rangle \tag{34}$$

**Combining iterations with and without restarts.** Summing up (33) and (34) for all $0 \leq t \leq T$,

$$\frac{\gamma}{2}\sum_{t=0}^{T}(f(\mathbf{x}_t) - f^\star) \leq \|\tilde{\mathbf{x}}_0 - \mathbf{x}^\star\|^2 - \|\tilde{\mathbf{x}}_{T+1} - \mathbf{x}^\star\|^2 + 2L\gamma^3\sum_{t=0}^{T}\left\|(\mathbf{b}_t - \mathbf{b}_{\lfloor\frac{t}{\tau}\rfloor\tau})^\top\mathbf{Z}\right\|^2$$

$$+ 2\gamma^2\sum_{k=1}^{\lfloor\frac{T}{\tau}\rfloor}\mathbb{E}\left\|(\mathbf{b}_{k\tau} - \mathbf{b}_{(k-1)\tau})^\top\mathbf{Z}\right\|^2 \underbrace{-2\gamma\sum_{k=1}^{\lfloor\frac{T}{\tau}\rfloor}\langle(\mathbf{b}_{k\tau} - \mathbf{b}_{(k-1)\tau})^\top\mathbf{Z}, \tilde{\mathbf{x}}_{k\tau-1} - \mathbf{x}^\star\rangle}_{:=S_1} \tag{35}$$

We now separately estimate the last sum $S_1$. We first divide it in pairs of two consecutive terms, and sum each pair separately. Lets denote $t = k\tau - 1$ for some $k$. Sum of two consecutive terms with indexes $t$ and $t - \tau$ is equal to

$$-2\gamma\langle(\mathbf{b}_{t+1} - \mathbf{b}_{t+1-\tau})^\top\mathbf{Z}, \tilde{\mathbf{x}}_t - \mathbf{x}^\star\rangle - 2\gamma\langle(\mathbf{b}_{t+1-\tau} - \mathbf{b}_{t+1-2\tau})^\top\mathbf{Z}, \tilde{\mathbf{x}}_{t-\tau} - \mathbf{x}^\star\rangle$$

$$= -2\gamma\langle(\mathbf{b}_{t+1} - \mathbf{b}_{t+1-\tau})^\top\mathbf{Z}, \tilde{\mathbf{x}}_t - \mathbf{x}^\star\rangle - 2\gamma\langle(\mathbf{b}_{t+1-\tau} - \mathbf{b}_{t+1-2\tau})^\top\mathbf{Z}, \tilde{\mathbf{x}}_t - \mathbf{x}^\star\rangle$$

$$- 2\gamma\langle(\mathbf{b}_{t+1-\tau} - \mathbf{b}_{t+1-2\tau})^\top\mathbf{Z}, \tilde{\mathbf{x}}_{t-\tau} - \tilde{\mathbf{x}}_t\rangle$$

$$= -2\gamma\langle(\mathbf{b}_{t+1} - \mathbf{b}_{t+1-2\tau})^\top\mathbf{Z}, \tilde{\mathbf{x}}_t - \mathbf{x}^\star\rangle - 2\gamma\langle(\mathbf{b}_{t+1-\tau} - \mathbf{b}_{t+1-2\tau})^\top\mathbf{Z}, \tilde{\mathbf{x}}_{t-\tau} - \tilde{\mathbf{x}}_t\rangle$$

Using update rules (14), it holds that $\tilde{\mathbf{x}}_t = \tilde{\mathbf{x}}_{t-\tau} - \gamma \sum_{j=t-\tau}^{t-1} \nabla f(\mathbf{x}_j) - \gamma(\mathbf{b}_{t+1-\tau} - \mathbf{b}_{t+1-2\tau})^\top \mathbf{Z}$, and thus

$$-2\gamma\langle(\mathbf{b}_{t+1-\tau} - \mathbf{b}_{t+1-2\tau})^\top \mathbf{Z}, \tilde{\mathbf{x}}_{t-\tau} - \tilde{\mathbf{x}}_t\rangle$$

$$= -2\gamma^2\langle(\mathbf{b}_{t+1-\tau} - \mathbf{b}_{t+1-2\tau})^\top \mathbf{Z}, \sum_{j=t-\tau}^{t-1} \nabla f(\mathbf{x}_j) + (\mathbf{b}_{t+1-\tau} - \mathbf{b}_{t+1-2\tau})^\top \mathbf{Z}\rangle$$

$$= \sum_{j=t-\tau}^{t-1} -2\gamma^2\langle(\mathbf{b}_{t+1-\tau} - \mathbf{b}_{t+1-2\tau})^\top \mathbf{Z}, \nabla f(\mathbf{x}_j)\rangle - 2\gamma^2 \left\|(\mathbf{b}_{t+1-\tau} - \mathbf{b}_{t+1-2\tau})^\top \mathbf{Z}\right\|^2$$

$$\overset{(25)}{\leq} \gamma^2\alpha\tau \left\|(\mathbf{b}_{t+1-\tau} - \mathbf{b}_{t+1-2\tau})^\top \mathbf{Z}\right\|^2 + \gamma^2\alpha^{-1} \sum_{j=t-\tau}^{t-1} \|\nabla f(\mathbf{x}_t)\|^2$$

$$- 2\gamma^2 \left\|(\mathbf{b}_{t+1-\tau} - \mathbf{b}_{t+1-2\tau})^\top \mathbf{Z}\right\|^2$$

$$\overset{\alpha=\frac{2}{\tau}}{\leq} \frac{\gamma^2\tau}{2} \sum_{j=t-\tau}^{t-1} \|\nabla f(\mathbf{x}_t)\|^2$$

Using these calculations, our original sum $S_1$ can be simplified as

$$S_1 \leq -2\gamma \sum_{k=1}^{\lfloor\frac{T}{2\tau}\rfloor} \langle(\mathbf{b}_{k\cdot 2\tau} - \mathbf{b}_{(k-1)2\tau})^\top \mathbf{Z}, \tilde{\mathbf{x}}_{k\cdot 2\tau-1} - \mathbf{x}^\star\rangle + \frac{\gamma^2\tau}{2} \sum_{t=0}^{\lfloor\frac{T}{\tau}\rfloor\tau-2} \|\nabla f(\mathbf{x}_t)\|^2$$

We reduced the sum of $\lfloor\frac{T}{\tau}\rfloor$ elements twice to the sum of the $\lfloor\frac{T}{2\tau}\rfloor$ elements. Continuing in similar way, we will need to have $\log_2\left(\lfloor\frac{T}{\tau}\rfloor\right)$ times until we reduce the original sum to just one element. Thus,

$$S_1 \leq -2\gamma\langle\mathbf{b}_{\lfloor\frac{T}{\tau}\rfloor\tau}^\top \mathbf{Z}, \tilde{\mathbf{x}}_{\lfloor\frac{T}{\tau}\rfloor\tau} - \mathbf{x}^\star\rangle + \frac{\gamma^2\tau}{2} \log_2\left(\left\lfloor\frac{T}{\tau}\right\rfloor\right) \sum_{t=0}^{\lfloor\frac{T}{\tau}\rfloor\tau-2} \|\nabla f(\mathbf{x}_t)\|^2$$

$$\overset{(25),\alpha=2}{\leq} \frac{1}{3} \left\|\tilde{\mathbf{x}}_{\lfloor\frac{T}{\tau}\rfloor\tau} - \mathbf{x}^\star\right\|^2 + 3\gamma^2 \left\|\mathbf{b}_{\lfloor\frac{T}{\tau}\rfloor\tau}^\top \mathbf{Z}\right\|^2 + \frac{\gamma^2\tau}{2} \log_2\left(\left\lfloor\frac{T}{\tau}\right\rfloor\right) \sum_{t=0}^{\lfloor\frac{T}{\tau}\rfloor\tau-2} \|\nabla f(\mathbf{x}_t)\|^2$$

We further transform the first term using the update rule (14)

$$\tilde{\mathbf{x}}_{T+1} = \tilde{\mathbf{x}}_{\lfloor\frac{T}{\tau}\rfloor\tau} - \gamma \sum_{j=\lfloor\frac{T}{\tau}\rfloor\tau}^{T} \nabla f(\mathbf{x}_j) = \tilde{\mathbf{x}}_{\lfloor\frac{T}{\tau}\rfloor\tau-1} - \gamma \sum_{j=\lfloor\frac{T}{\tau}\rfloor\tau-1}^{T} \nabla f(\mathbf{x}_j) - \gamma \left(\mathbf{b}_{\lfloor\frac{T}{\tau}\rfloor\tau} - \mathbf{b}_{(\lfloor\frac{T}{\tau}\rfloor-1)\tau}\right)^\top \mathbf{Z}$$

Thus,

$$\frac{1}{3} \left\|\tilde{\mathbf{x}}_{\lfloor\frac{T}{\tau}\rfloor\tau} - \mathbf{x}^\star\right\|^2 \leq \|\tilde{\mathbf{x}}_{T+1} - \mathbf{x}^\star\|^2 + \gamma^2\tau \sum_{j=\lfloor\frac{T}{\tau}\rfloor\tau-1}^{T} \|\nabla f(\mathbf{x}_j)\|^2 + \gamma^2 \left\|\left(\mathbf{b}_{\lfloor\frac{T}{\tau}\rfloor\tau} - \mathbf{b}_{(\lfloor\frac{T}{\tau}\rfloor-1)\tau}\right)^\top \mathbf{Z}\right\|^2$$

And thus,

$$S_1 \leq \|\tilde{\mathbf{x}}_{T+1} - \mathbf{x}^\star\|^2 + 3\gamma^2 \left\|\mathbf{b}_{\lfloor\frac{T}{\tau}\rfloor\tau}^\top \mathbf{Z}\right\|^2 + \gamma^2\tau \log_2\left(\left\lfloor\frac{T}{\tau}\right\rfloor\right) \sum_{t=0}^{T} \|\nabla f(\mathbf{x}_t)\|^2$$

$$+ \gamma^2 \left\|\left(\mathbf{b}_{\lfloor\frac{T}{\tau}\rfloor\tau} - \mathbf{b}_{(\lfloor\frac{T}{\tau}\rfloor-1)\tau}\right)^\top \mathbf{Z}\right\|^2$$

Choosing $\tau = \frac{1}{8L\gamma\log_2(T)}$ ensures that $\gamma^2\tau \log_2\left(\lfloor\frac{T}{\tau}\rfloor\right) \leq \frac{\gamma}{8L}$. Putting these calculations back into (35), we get that

$$\frac{\gamma}{2} \sum_{t=0}^{T}(f(\mathbf{x}_t) - f^\star) \leq \|\tilde{\mathbf{x}}_0 - \mathbf{x}^\star\|^2 + 3L\gamma^3 \sum_{t=0}^{T} \left\|(\mathbf{b}_{t-1} - \mathbf{b}_{\lfloor\frac{t}{\tau}\rfloor\tau})^\top \mathbf{Z}\right\|^2$$

$$+ 3\gamma^2 \sum_{k=1}^{\lfloor\frac{T}{\tau}\rfloor} \mathbb{E} \left\|(\mathbf{b}_{k\tau} - \mathbf{b}_{(k-1)\tau})^\top \mathbf{Z}\right\|^2 + \frac{\gamma}{8L} \sum_{t=0}^{T} \|\nabla f(\mathbf{x}_t)\|^2 + 3\gamma^2 \left\|\mathbf{b}_{\lfloor\frac{T}{\tau}\rfloor\tau}^\top \mathbf{Z}\right\|^2$$

Using (32), we can further simplify

$$\frac{\gamma}{4}\sum_{t=0}^{T}(f(\mathbf{x}_t) - f^\star) \leq 3\gamma^2\Big(L\gamma\sum_{t=0}^{T}\left\|(\mathbf{b}_t - \mathbf{b}_{\lfloor\frac{t}{\tau}\rfloor\tau})^\top\mathbf{Z}\right\|^2 + \sum_{k=1}^{\lfloor\frac{T}{\tau}\rfloor}\mathbb{E}\left\|(\mathbf{b}_{k\tau} - \mathbf{b}_{(k-1)\tau})^\top\mathbf{Z}\right\|^2$$

$$+ \left\|\mathbf{b}_{\lfloor\frac{T}{\tau}\rfloor\tau}^\top\mathbf{Z}\right\|^2\Big) + \|\mathbf{x}_0 - \mathbf{x}^\star\|^2$$

## D  Convergence of Anti-PGD

Here we discuss the convergence of the Anti-PGD method, introduced in Example 3.2.

Since $\tilde{\mathbf{x}}_{t+1} = \tilde{\mathbf{x}}_t - \gamma\nabla f(\mathbf{x}_t)$, for some point $\mathbf{x}^\star$ that satisfies $\nabla f(\mathbf{x}^\star) = 0$,

$$\|\tilde{\mathbf{x}}_{t+1} - \mathbf{x}^\star\|^2 = \|\tilde{\mathbf{x}}_t - \mathbf{x}^\star\|^2 - 2\gamma\langle\nabla f(\mathbf{x}_t), \mathbf{x}_t - \mathbf{x}^\star\rangle + \gamma^2\|\nabla f(\mathbf{x}_t)\|^2 + 2\gamma\langle\nabla f(\mathbf{x}_t), \mathbf{x}_t - \tilde{\mathbf{x}}_t\rangle$$

$$\overset{(32),(23)}{\leq} \|\tilde{\mathbf{x}}_t - \mathbf{x}^\star\|^2 - 2\gamma(1 - L\gamma)(f(\mathbf{x}_t) - f^\star) + 2\gamma\langle\nabla f(\mathbf{x}_t), \mathbf{x}_t - \tilde{\mathbf{x}}_t\rangle$$

We estimate the last term separately

$$2\langle\nabla f(\mathbf{x}_t), \mathbf{x}_t - \tilde{\mathbf{x}}_t\rangle \overset{(25),\alpha=2L}{\leq} \frac{1}{2L}\|\nabla f(\mathbf{x}_t)\|^2 + 2L\|\mathbf{x}_t - \tilde{\mathbf{x}}_t\|^2 \overset{(32)}{\leq} (f(\mathbf{x}_t) - f^\star) + 2L\|\mathbf{x}_t - \tilde{\mathbf{x}}_t\|^2$$

Thus,

$$\|\tilde{\mathbf{x}}_{t+1} - \mathbf{x}^\star\|^2 \leq \|\tilde{\mathbf{x}}_t - \mathbf{x}^\star\|^2 - \gamma(1 - 2L\gamma)(f(\mathbf{x}_t) - f^\star) + 2L\gamma\|\mathbf{x}_t - \tilde{\mathbf{x}}_t\|^2$$

$$\overset{\gamma<\frac{1}{4L},(20)}{\leq} \|\tilde{\mathbf{x}}_t - \mathbf{x}^\star\|^2 - \frac{\gamma}{2}(f(\mathbf{x}_t) - f^\star) + 2L\gamma\|\mathbf{x}_t - \tilde{\mathbf{x}}_t\|^2$$

## E  Noise Lower Bound

We consider function $f(\mathbf{x}) = \frac{L}{2}\|\mathbf{x}\|^2$ that is convex and $L$-smooth, and we are running algorithm (8) with constant stepsize $\gamma$, and we consider the two cases of $\mathbf{B} = \mathbf{S}$ and $\mathbf{B} = \mathbf{I}$.

### E.1  PGD

This corresponds to Example 3.1. We will prove the lower bound on the noise term under the condition that $T$ is large enough, i.e. $T \geq \frac{\log 2}{\eta L}$.

Since $\nabla f(\mathbf{x}) = L\mathbf{x}$, the algorithm (8) takes a form

$$\mathbf{x}_{t+1} = (1 - \gamma L)\mathbf{x}_t - \gamma\mathbf{z}_{t+1}$$

Thus, since $\mathbf{x}^\star = 0$,

$$\mathbb{E}\|\mathbf{x}_{t+1}\|^2 = \mathbb{E}\|(1 - \gamma L)\mathbf{x}_t - \gamma\mathbf{z}_{t+1}\|^2 = (1 - \gamma L)^2\|\mathbf{x}_t\|^2 + \gamma^2\sigma^2$$

$$= (1 - \gamma L)^{2(t+1)}\|\mathbf{x}_0\|^2 + \gamma^2\sigma^2\sum_{j=0}^{t}(1 - \gamma L)^{2j}$$

due to the unbiasedness and independence of $\mathbf{z}_t$. We can exactly calculate the sum of this geometric series

$$\sum_{j=0}^{T-1}(1 - \gamma L)^{2j} = \frac{1 - (1 - \gamma L)^{2T}}{1 - (1 - \gamma L)^2} = \frac{1 - (1 - 2\gamma L)^{2T}}{2\gamma L - \gamma^2 L^2} \geq \frac{1}{4\gamma L}$$

where at the last step we used that $\gamma^2 L^2 > 0$ and that $T \geq \frac{\log 2}{\gamma L}$.

And thus the function values are larger than

$$f(\mathbf{x}_T) - f^\star = \frac{L}{2}\|\mathbf{x}_T\|^2 \geq \frac{L}{2}(1 - \gamma L)^{2(t+1)}\|\mathbf{x}_0\|^2 + \frac{1}{8}\gamma\sigma^2$$

This shows that the noise term in (11) cannot be improved.

## E.2 Anti-PGD

This corresponds to Example 3.2. Since $\nabla f(\mathbf{x}) = L\mathbf{x}$, the algorithm (8) for Anti-PGD noise takes a form

$$
\begin{aligned}
\mathbf{x}_{t+1} &= (1-\gamma L)\mathbf{x}_t - \gamma \mathbf{z}_{t+1} + \gamma \mathbf{z}_t \\
&= (1-\gamma L)^2 \mathbf{x}_{t-1} - (1-\gamma L)\gamma \mathbf{z}_t + (1-\gamma L)\gamma \mathbf{z}_{t-1} - \gamma \mathbf{z}_{t+1} + \gamma \mathbf{z}_t \\
&= (1-\gamma L)^2 \mathbf{x}_{t-1} + (1-\gamma L)\gamma \mathbf{z}_{t-1} - \gamma \mathbf{z}_{t+1} + \gamma^2 L \mathbf{z}_t \\
&= (1-\gamma L)^{t+1} \mathbf{x}_0 + (1-\gamma L)^t \gamma \mathbf{z}_1 + \gamma^2 L \sum_{j=1}^{t-1} (1-\gamma L)^{t-j} \mathbf{z}_{j+1} - \gamma \mathbf{z}_{t+1}
\end{aligned}
$$

Thus,

$$
\mathbb{E} \left\| \mathbf{x}_T \right\|^2 = (1-\gamma L)^{2T} \left\| \mathbf{x}_0 \right\|^2 + (1-\gamma L)^{2(T-1)} \gamma^2 \, \mathbb{E} \left\| \mathbf{z}_1 \right\|^2 + \gamma^4 L^2 \sum_{j=1}^{T-2} (1-\gamma L)^{2(T-1-j)} \, \mathbb{E} \left\| \mathbf{z}_{j+1} \right\|^2
$$

$$
+ \gamma^2 \, \mathbb{E} \left\| \mathbf{z}_{t+1} \right\|^2 \geq (1-\gamma L)^{2T} \left\| \mathbf{x}_0 \right\|^2 + \gamma^2 \sigma^2
$$

Thus the function values are larger than

$$
f(\mathbf{x}_T) - f^\star = \frac{L}{2} \left\| \mathbf{x}_T \right\|^2 \geq (1-\gamma L)^{2T} \left\| \mathbf{x}_0 \right\|^2 + \frac{L}{2} \gamma^2 \sigma^2
$$

This proves that the noise term in (12) cannot be improved.

## E.3 Virtual Sequence for PGD

In this section we show that for the PGD algorithm, virtual sequences $\tilde{\mathbf{x}}_t$ that are defined in (13) cannot give a tight convergence result.

Since $\tilde{\mathbf{x}}_{t+1} = \tilde{\mathbf{x}}_t - \gamma \nabla f(\mathbf{x}_t)$, and $\nabla f(\mathbf{x}_t) = L\mathbf{x}_t$ we get

$$
\tilde{\mathbf{x}}_{t+1} = (1-\gamma L)\tilde{\mathbf{x}}_t + \gamma L(\tilde{\mathbf{x}}_t - \mathbf{x}_t) = (1-\gamma L)\tilde{\mathbf{x}}_t - \gamma^2 L \sum_{j=0}^{t} \mathbf{z}_j
$$

where the last equality is since $\tilde{\mathbf{x}}_t - \mathbf{x}_t = -\gamma \sum_{j=1}^{t} \mathbf{z}_j$. Unrolling,

$$
\tilde{\mathbf{x}}_{t+1} = (1-\gamma L)\tilde{\mathbf{x}}_t + \gamma L(\tilde{\mathbf{x}}_t - \mathbf{x}_t) = (1-\gamma L)^{t+1} \tilde{\mathbf{x}}_0 - \gamma^2 L \sum_{j=1}^{t} \mathbf{z}_j \sum_{i=0}^{j} (1-\gamma L)^i
$$

Thus the norm

$$
\mathbb{E} \left\| \tilde{\mathbf{x}}_T \right\|^2 = (1-\gamma L)^{2T} \left\| \mathbf{x}_0 \right\|^2 + \gamma^4 L^2 \sum_{j=1}^{T-1} \left[ \sum_{i=0}^{j} (1-\gamma L)^i \right]^2 \sigma^2
$$

We can calculate exactly the inner sum as

$$
\sum_{i=0}^{j} (1-\gamma L)^i = \frac{1 - (1-\gamma L)^j}{\gamma L}
$$

and thus

$$
\mathbb{E} \left\| \tilde{\mathbf{x}}_T \right\|^2 = (1-\gamma L)^{2T} \left\| \mathbf{x}_0 \right\|^2 + \gamma^2 \sum_{j=1}^{T-1} \left[ 1 - (1-\gamma L)^j \right]^2 \sigma^2 \geq \gamma^2 \sum_{j=\frac{T}{2}}^{T-1} \left[ 1 - 2(1-\gamma L)^j \right] \sigma^2
$$

It is left to note that for $T$ sufficiently large, $T \geq 2\frac{\log 4}{\gamma L}$, it holds that $(1-\gamma L)^{T/2} \leq \frac{1}{4}$ and thus $\left[ 1 - 2(1-\gamma L)^j \right] \geq \frac{1}{2}$. Using this, we arrive

$$
\mathbb{E} \left\| \tilde{\mathbf{x}}_T \right\|^2 \geq \gamma^2 \sigma^2 \frac{T}{4}
$$

and this the function value $f(\tilde{\mathbf{x}}_T) \geq L\gamma^2 \sigma^2 \frac{T}{8}$.

# F  Difficulties in Deriving a Unified Analysis

In this section we explain the difficulties in unifying theoretical analysis using existing proof techniques described in the main text. In particular analysis through the real iterates $\mathbf{x}_t$ can give good convergence guarantees only for PGD, but not Anti-PGD, and vise versa, analysis through the virtual iterates $\tilde{\mathbf{x}}_t$ can give a good convergence guarantee for Anti-PGD but not for PGD.

Directly analyzing Anti-PGD using the actual iterates $\mathbf{x}_t$ of (7), we only get a convergence rate of

$$\sum_{t=0}^{T} \frac{\mathbb{E}\left[f(\mathbf{x}_t) - f^\star\right]}{T+1} \leq \mathcal{O}\left(\frac{\|\mathbf{x}_0 - \mathbf{x}^\star\|^2}{\gamma T} + \gamma \sigma^2\right).$$

Note that this is strictly worse than the Anti-PGD rate in (12). While we do not see any fundamental limit to analysing Anti-PGD directly through its iterates $\mathbf{x}_t$, we do not know of how to do so in a way that recovers the rate in (12).

On the other hand, applying the perturbed iterate analysis (via the virtual sequence $\tilde{\mathbf{x}}_t$ produced by (7) when $\mathbf{Z} = \mathbf{0}$) to PGD, we only get a convergence rate of

$$\sum_{t=0}^{T} \frac{\mathbb{E}\left[f(\mathbf{x}_t) - f^\star\right]}{T+1} \leq \mathcal{O}\left(\frac{\|\mathbf{x}_0 - \mathbf{x}^\star\|^2}{\gamma T} + LT\gamma^2\sigma^2\right).$$

This rate is strictly worse than the rate derived through a virtual sequence in (11) when $\gamma > 1/LT$. As we detail in Appendix E, this bound is actually a tight upper bound for the convergence of the virtual sequence $f(\tilde{\mathbf{x}}_t)$. However, the real sequence $\mathbf{x}_t$ converges faster than this according to (11). In short, while one can use the virtual sequence $\tilde{\mathbf{x}}_t$ to effectively analyze anti-correlated noise, such techniques do not directly yield a tight analysis of PGD.

# G  Applying Theorem 4.7 to special cases

**PGD.**  In this case, $\mathbf{B} = \mathbf{S}$ (Example 2.1), so if $i - j \leq \tau$ then $\|\mathbf{b}_i - \mathbf{b}_j\|^2 \leq \tau$. The noise term in the convergence rate of Theorem 4.7 is therefore upper bounded by

$$\frac{\sigma^2}{TL\tau}\left[\frac{1}{\tau}\sum_{t=1}^{T}\tau + \sum_{\substack{1 \leq t \leq T \\ t = 0 \bmod \tau}}\tau + T\right] = \tilde{\mathcal{O}}\left(\frac{\sigma^2}{L\tau}\right) = \tilde{\mathcal{O}}\left(\gamma\sigma^2\right)$$

This matches the tight convergence rate in Proposition 4.4.

**Anti-PGD.**  Since $\mathbf{B} = \mathbf{I}$, for any rows $\mathbf{b}_i$, $\mathbf{b}_j$, $\|\mathbf{b}_i - \mathbf{b}_j\|^2 \leq 2$. Thus, the noise term in the convergence rate of Theorem 4.7 is upper bounded by

$$\frac{\sigma^2}{TL\tau}\left[\frac{1}{\tau}\sum_{t=1}^{T}2 + \sum_{\substack{1 \leq t \leq T \\ t = 0 \bmod \tau}}2 + 1\right] = \tilde{\mathcal{O}}\left(\frac{\sigma^2}{L\tau^2}\right) = \tilde{\mathcal{O}}\left(L\gamma^2\sigma^2\right)$$

where we used $\tau = \tilde{\mathcal{O}}(1/L\gamma)$. This recovers the tight convergence rate in Proposition 4.5.

# H  Experiments

In this section we provide the complete experimental details for the experiments in Section 6, as well as additional experiments on the Stack Overflow dataset.

## H.1  Experiments with Quadratic Functions

We study *random quadratic* function $f(\mathbf{x}) = \frac{1}{2}\|\mathbf{Ax} - \mathbf{b}\|^2$ to be able to precisely control the smoothness constant $L$ that appears in our theoretical analysis. In particular, we set the spectrum of $\mathbf{A} \in \mathbb{R}^{100 \times 100}$ to have the values to be linearly distributed between $\lambda_{\min} = 0$ and $\lambda_{\max} = \sqrt{L}$,

| Dataset | MNIST | CIFAR-10 | StackOverflow |
|---|---|---|---|
| Train Records | 60,000 | 50,000 | 135,818,730 |
| Test Records | 10,000 | 10,000 | 16,586,035 |
| Dimensionality | 784 | 3,072 | 200,000 |
| Classes | 10 | 10 | 10,000 |
| Model | Logistic | CNN | LSTM |
| Privacy Unit | Example | Example | User |
| Parameters | 7,056 | 550,570 | 4,050,748 |
| Learning Setting | Centralized | Centralized | Federated |

Table 1: Summary of datasets and associated problems considered in this empirical evaluation.

and we randomly shift the axis by unitary transformation. We calculate the unitary transformation by the SVD of a random matrix $\mathbf{D}$ with every element $d_{ij} \in \mathcal{N}(0, 1)$. Lets $\mathbf{D} = \mathbf{U}_D \mathbf{\Lambda}_D \mathbf{V}_D$ be the SVD decomposition, and let $\mathbf{\Lambda}_A = \mathrm{diag}(\lambda_{\max}, \dots, \lambda_{\min})$ is the matrix with the desired spectrum (between $\lambda_{\min} = 0$ and $\lambda_{\max} = \sqrt{L}$). We calculate the matrix $\mathbf{A}$ as $\mathbf{A} = \mathbf{U}_D \mathbf{\Lambda}_A \mathbf{V}_D$. We also randomly sample the shift $\mathbf{b} \in \mathcal{N}(0, \mathbf{I})$, $\mathbf{b} \in \mathbb{R}^{100}$.

We note that such quadratic function $f$ is $L$-smooth and convex. We fix the number of iterations $T$ to 5000, and the variance of the noise $\sigma$ is equal to 20.

In these experiments we aim to compare DP-MF, and our proposed DP-MF$^+$ methods under varying hyperparameter settings. We fix the smoothness $L = 10$, and we vary the learning rate $\gamma$ over the logarithmic grid between $10^{-4}$ and 1, and we further select the region of learning rates around the optimal $\gamma$. We also tune parameter $\tau$ in DP-MF$^+$ over the grid $\{1, 2, 10, 50, 100, 200, 500, 1000, 5000\}$.

## H.2 Practical DP Training Experiments

**Datasets and tasks.** Table 1 summarizes the datasets and problems used in our empirical evaluation. For the MNIST dataset, light preprocessing is done so the $28 \times 28$ input images are flattened to size 784 vectors and normalized so entries lie in the range $[0, 1]$. For the CIFAR-10 and Stack Overflow datasets, the experimental setup including data preprocessing follows exactly from Denisov et al. [10] and Choquette-Choo et al. [7].

**Metrics.** For each dataset, mechanism, and privacy parameter, we run the mechanism for multiple trials and report the test set accuracy of the final iterate. We compute the mean and standard error of the reported test set accuracies.

**MNIST, logistic regression.** For MNIST we train a logistic regression model to predict image labels. All mechanisms train for $T = 2048$ iterations and either 1 or 16 epochs, corresponding to batch sizes of 29 and 469 respectively.[4] We vary $\varepsilon$ over $\{0.01, 0.1, \dots, 100\}$ and fix $\delta = 10^{-6}$. We fix the clipping threshold at $1.0$ and the learning rate at $0.5$. We run each experiment for 5 trials, and plot the mean test set accuracy along with error bars indicating the standard error of the estimate.

**CIFAR-10, CNN.** For CIFAR-10, we follow the experimental setup from [7] and train a CNN model to predict image labels. Specifically, we train all mechanisms for 20 epochs and $T = 2000$ iterations, which corresponds to a batch size of 500.[5] We consider $\varepsilon = 1, 2, 4, 8, 16, 32$ and set $\delta = 10^{-6}$. We tune the learning rate non-privately for each method and $\varepsilon$ by running a single trial with a fixed random seed and choosing the one which achieved the lowest training error. For each value of $\varepsilon$, we use the tuned learning rate and run 12 new trials with different random seeds, and record the test set accuracy at the end of training.

**Stack Overflow, LSTM.** We follow the experimental setup of Denisov et al. [10], and train a next-word prediction LSTM model on the Stack Overflow dataset [3]. We train each mechanism

---

[4]In practice, one often trains small-scale models for many epochs, perhaps even using full-batch gradients, to improve the privacy/utility trade-off (at the cost of increased computation). We are interested in the *relative* performance for a fixed computation budget, so we train for a small number of epochs.

[5]While Choquette-Choo et al. [7] use momentum and learning rate decay, we omit the use of such techniques as they are orthogonal to our theoretical results.

| Noise Multiplier | DP-MF | DP-MF$^+$($\tau = 2048$) |
|---|---|---|
| 0.341 | $24.63 \pm 0.06$ | $24.58 \pm 0.12$ |
| 0.682 | $23.76 \pm 0.14$ | $23.73 \pm 0.16$ |
| 1.364 | $22.54 \pm 0.11$ | $22.44 \pm 0.08$ |
| 2.728 | $11.51 \pm 12.71$ | $10.42 \pm 13.05$ |
| 5.456 | $0.03 \pm 0.02$ | $0.05 \pm 0.06$ |

Table 2: Comparison of test set accuracies on the Stack Overflow next word prediction task between DP-MF and DP-MF$^+$.

for 1 epoch and $2048$ iterations, which corresponds to about $167$ clients per round, each holding an average of $\approx 400$ records. We vary the hyper-parameters according to prior work and run 2 trials for each hyper-parameter setting. We report results for the best hyper-parameters setting of each mechanism. We use federated averaging instead of gradient descent. Additionally, to be consistent with the prior work and to test if our proposed factorizations are compatible with the other types of workloads, we use momentum and learning rate decay. Although the $\mathbf{C}$ matrix was optimized for the Prefix workload, $\mathbf{A} = \mathbf{S}$, it is applied to a variant $\mathbf{A} = \mathbf{S}'$ that incorporates momentum and learning rate decay by setting $\mathbf{B} = \mathbf{S}'\mathbf{C}^{-1}$. More details of how DP-MF and DP-MF$^+$ apply to this setting are available in Denisov et al. [10].

The results are shown in Table 2 for varying the noise multiplier, which corresponds to values of $\varepsilon$ are equal to $\{17.65, 7.6, 3.44, 1.61, 0.76\}$. We see no significant difference between DP-MF and DP-MF$^+$, as the small differences in performance are within the statistical bounds one would expect if they had identical means. At larger noise multipliers, both DP-MF and DP-MF$^+$ exhibit learning instabilities.