# OpenReview forum: "Gradient Descent with Linearly Correlated Noise: Theory and Applications to Differential Privacy"
_NeurIPS.cc/2023/Conference — NeurIPS 2023 poster_

### Official Review · Reviewer_xFGx · 2023-07-06

**Soundness:** 4 excellent
**Presentation:** 4 excellent
**Contribution:** 3 good
**Rating:** 6
**Confidence:** 4

**Summary:**

The authors propose a new analysis method that unifies the analysis of noisy gradient descent-based algorithms, where the noise is added to the matrix factorization, which provides a tight analysis of both the algorithms simultaneously, and hence would give the analysis for convergences for a general class of such matrix-factorization based gradient descent algorithms. Based on this new analysis, the authors also propose a new matrix factorization objective to further tighten the upper bound and empirically show it.

**Strengths:**

- The method of analysis is based on a very simple yet essential idea and it opens the possibilities of a unified method of analysis for a good class of algorithms which are based on gradient descent algorithms based on matrix factorization.
- The transition from proposing a tighter bound to a new matrix factorization objective shows powerful empirical applications of this class of algorithms to private optimization.
- The paper is theoretically very sound.
- Evaluation done on the paper is simple and easy to understand and evaluation has been done not only with respect to utility but also to visualize the trend that the gradients follow across training.

**Weaknesses:**

- The novelty and the reach of this problem are limited to a very specific class of algorithms on a practical scale.
- Although the authors have already addressed this in their future work, it is still important to note that the average case analysis for differentially private algorithms might not be a correct metric to judge the performance of the model since the final iterate of the algorithm is returned.

**Questions:**

- Can the authors elaborate on what they exactly want to denote using Figure 1 and maybe provide a more clear diagram for it so that the process can be understood better?

**Limitations:**

Authors have mentioned most of the limitations in their future work and additional points have been discussed above in the weakness section (novely and final iterate analysis).

---

> ### Author Rebuttal · Authors · 2023-08-09
>
> Thank you very much for your time and positive review! Below we address the raised points.
>
> ## Weaknesses
>
> 1. While our analysis is restricted to the specific class of algorithms, this class of algorithms capture significant advances in differentially private optimization which were recently shown to strictly generalize (and dominate) the major methods of differentially private optimization, practically improving over a popular and widely-used DP-SGD algorithm [1].
>
>     Our work analyzes the dynamics of a general matrix mechanism applied to model training with gradient descent, and is therefore applicable to any of the matrix-based mechanisms recently designed for improved scalability. To the best of our knowledge, none of the related work has moved beyond the Frobenius-norm formulation of error which we show to fail to be predictive of optimization performance in a variety of cases. This is to say: our work inherits the applicability, algorithmic and scalability improvements pursued in the rest of the literature, but addresses a fundamentally orthogonal problem. For this reason, we expect the domain of application of the analysis presented here to continue to grow in the near future.
>
> [1] Choquette-Choo et al,  (Amplified) Banded Matrix Factorization: A unified approach to private training.
>
> ## Questions
>
> 1. In Figure 1 we aimed to illustrate the workflow of MF-DP-FTRL algorithms. There are two underlying optimization problems: a one-time matrix factorization problem (left) that is performed to obtain a noise correlation pattern B, depending on the workload matrix A, and a factorization objective; and (left) an ERM minimization that is data and loss-dependent, performed using the found noise correlation pattern B. Our work focuses on understanding the (right) part of this diagram: specifically, we aim to understand which factorization objective one should use for a one-time offline matrix factorization problem.
> We will make our diagram more clear in the next version. In particular we will explain the connection of our diagram to the MF-DP-FTRL algorithm in (4).

---

> > ### Comment · Reviewer_xFGx · 2023-08-12
> > **Response to Authors**
> >
> > Thanks to the Authors for addressing the questions. I do not have further questions.

---

### Official Review · Reviewer_2HBd · 2023-07-07

**Soundness:** 3 good
**Presentation:** 3 good
**Contribution:** 3 good
**Rating:** 7
**Confidence:** 2

**Summary:**

This paper conducted a theoretical study of gradient descent with linearly correlated noise, with strong motivation from the theory and applications of differential privacy (in particular, the popular DP-SGD algorithm).

The main contribution is a new convergence analysis based on the idea of "restart iterations". The paper argues that it unifies two previous  "incompatible" analyses for two different noisy gradient descent algorithms, namely perturbed gradient descent (PGD) and anti-PGD.

Inspired by the new analysis, this paper proposes a new matrix factorization method for noisy SGD. The matrix factorization coordinates noises across different iterations of gradient descent. The new factorization refines a previous method. Experiments show that it compares favorably to prior works.


**Strengths:**

* The paper is very well-written and looks both educational and professional.
* It tackles an important and popular question of optimization and private learning. The "restart iteration" analysis looks nice. However, I don't work on this area myself, it is hard for me to evaluate the novelty of the new analysis.

**Weaknesses:**

* I am slightly dissatisfied with the hyper-parameter "\tau" required in your new algorithm and analysis (more on this in the question section).

**Questions:**

* If possible, it would be nice to discuss why the two analyses for PGD and anti-PGD are incompatible. Also, I am curious if the two different proofs directly inspire your analysis.

* Your paper shows that the "restart" in analysis and algorithm design does help. However, could you provide some intuition on why "restart" is beneficial in both theory and applications? Do you have evidence that this method yields tight analysis?

* Are there good heuristics to find an appropriate "tau" in practice? Is there a way to come up with an algorithm that is equally good, but does not require the knowledge of the hyperparameter?

**Limitations:**

The limitations and future directions are discussed in the paper.

---

> ### Author Rebuttal · Authors · 2023-08-09
>
> ## Questions
>
> 1. Thanks for your question. Due to the space limitations, we discussed why the two analyses for PGD and anti-PGD are incompatible in the appendix F, where we showed that
> - using the real iterate sequence $x_t$ we cannot prove the tight convergence bound of Anti-PGD (12).
> - using the virtual iterate sequence $\tilde x_t$, we cannot prove the tight convergence bound of PGD. We also show in Appendix E.3. that for the PGD algorithm the virtual sequence $\tilde x_t$ is provably converging slower than the real iterate sequence $x_t$, making it fundamentally impossible to get tight rates of PGD through $\tilde x_t$.
>
>     While we do not see any fundamental limitations of analyzing Anti-PGD using the real sequence $x_t$ we are unaware of how to prove a tight rate (12) using the real sequence $x_t$.
>
>     The two different proofs directly inspired our analysis: our approach combines virtual and real iterates through the restart trick. Every tau iterations the virtual iterate $\tilde x_t$ is restarted to the real iterate $x_t$, allowing for the tight analysis of both algorithms simultaneously.
>
> 2. Our analysis is tighter than the prior rate in (5). Indeed, it holds that $|| \Lambda_{\tau} B||^2_F \leq 4 ||B||^2_F$, making our rates in Theorem 4.7. strictly tighter than that in (5). Moreover, our derived convergence rate in Theorems 4.6. and 4.7. is tight for some notable special cases of matrix B such as PGD, Anti-PGD and Chess-PGD (discussed in Appendix A.2), while the prior convergence rate based on $||B||^2_F$ cannot be tight for these cases (see discussion in lines 130-137, as well as in Appendix A). Therefore, minimizing a tighter bound based on $||\Lambda_{\tau} B||^2_F$ (under the fixed privacy) should directly transfer to the better optimization properties than minimizing a bound based on a loose bound $||B||^2_F$.
>
> 3. While in our experiments we found that setting $\tau = T$ gave the best results in practice, our experiments are relatively small-scale. We envision that for the larger scale experiments, one should set $\tau$ as large so that computing factorization of matrix $\tau \times \tau$ is still computationally feasible.
>
>     On the other hand, our experiments in Figure 2 revealed a mismatch between the average and the last iterate behavior. We believe that analyzing our method for the last-iterate behavior can help to understand which $\tau$ we should pick to achieve the best performance in practice. This question is out of scope of our work, where we focused on understanding the convergence properties for the average iterate.

---

> > ### Comment · Reviewer_2HBd · 2023-08-18
> > **Thank you**
> >
> > Thanks to the authors for answering my questions!

---

### Official Review · Reviewer_WD4o · 2023-07-21

**Soundness:** 4 excellent
**Presentation:** 4 excellent
**Contribution:** 4 excellent
**Rating:** 8
**Confidence:** 5

**Summary:**

This paper studies (stochastic) gradient descent with linearly correlated noise. The paper builds upon recent results on (MF-)DP-FTRL. It highlights the limitations of this methods, proposes a restarting trick to improve it, and derive the corresponding analysis. Interestingly, the proposed analysis gives a unified analysis of PGD and Anti-PGD, that are difficult to analyze together otherwise. Theoretical results are confirmed by numerical experiments.


**Strengths:**

Overall, I think this paper is a nice paper, that perfectly fits the conference.
1. The proposed method identifies the limitations of an (important) existing method and proposes a clever trick to overcome them.
2. Corresponding theory is derived in multiple settings (convex and non-convex), and is confirmed by numerical experiments.
3. Overall, the paper is very pleasant to read (even the appendix), with a clear introduction and clear motivations, and the math seems to be correct.


**Weaknesses:**

In my opinion, this paper does not have major weaknesses. I only have a few minor remarks.
1. The method is claimed to work for DP-SGD, but the theory is only derived with full-batch gradients. This is a bit unfortunate since DP-FTRL-like methods aim at providing stochastic DP algorithms that do not rely on amplification.
2. The proposed method seems a bit "handmade": it seems there is something deeper behind it.
3. Example 3.3 is a bit crude and could benefit from a little more details (on the choice of the pseudo-inverse and the fact that B and C are not square).


**Questions:**

1. The restarting trick essentially states that not all iterates should have correlated noise. Do you think this is a fundamental property of the method or simply a caveat of the analysis?
2. How does the restarting trick affect numerical performances in terms of time of execution? It seems that solving $OPT_F(\Lambda_\tau A)$ could be easier due to the structure of $\Lambda_\tau$: is it the case?


**Limitations:**

Yes, authors pointed that their analysis does not encompass clipping, and only works for averaged iterate. Some other research directions/limitations have also been mentioned in the conclusion.

---

> ### Author Rebuttal · Authors · 2023-08-09
>
> We would like to thank the reviewer for their time and their positive assessment of our paper.
>
> ## Weaknesses:
> 1. In this work we aimed to find a simple theoretical model for which it would be interesting to study the effect of linearly correlated noise on the optimization dynamics. Since the stochasticity in the gradients is not directly linked to the linearly correlated noise, for clarity we decided to omit it from the theoretical analysis. Our model serves as a proxy to the real experimental setup. We verified the effectiveness of our proposed method experimentally, justifying the simplifications made.
>
> 3. Thanks for your comment. We excluded the details due to space limitations. Upon your suggestion, we will add more explanations of Example 3.3. in the next version in the appendix or in the main text if space permits.
>
> ## Questions:
> Thanks for your interesting questions!
>
> 1. We believe that there is something deeper going on. Our analysis was motivated by the intuition that not all the iterates should have correlated noise: intuitively, if the noise was added large number $T_0$ iterations ago, the iterate $x_t$ should move sufficiently far from $x_{t - T_0}$ making the old noise added at iteration $t - T_0$ to become effectively uncorrelated. However, this is only an intuitive explanation, and more investigation is needed to understand this question in detail.
>
> 2. Thanks for your question. Because of the repetitive pattern of matrix $\Lambda_{\tau}$ (See Appendix B), we need to perform optimization only for the block of $\tau \times \tau$ instead of $T \times T$ matrix $A$, repeating the found correlation pattern $B_{\tau \times \tau}$ $T / \tau$ times. See also our attached PDF to the reviewer dQeP for illustration of the repetitive pattern in $B$. We will include these details in the next version.

---

> > ### Comment · Reviewer_WD4o · 2023-08-13
> >
> > Thank you for your answer and for the additional precisions. I don't have further comments.

---

### Official Review · Reviewer_ptMd · 2023-07-22

**Soundness:** 3 good
**Presentation:** 3 good
**Contribution:** 3 good
**Rating:** 5
**Confidence:** 2

**Summary:**

This paper develops many techniques to analyze gradient descent convergence with linearly correlated noise, a setting motivated by many DP algorithms. The work derives tighter bounds, and use the resulting insights to motivate DP-MF and DP-MF+.

**Strengths:**

- The paper is structured well, and is very notationally precise for the most part.
- The theoretical results are very involved, and I particularly appreciate the authors including several noteworthy special cases.

**Weaknesses:**

Clarity:
- This work is not very self contained, and it seems like the authors expect the reader to be very familiar with Denisov et al. and Choquette-Choo et al. This makes the work a bit difficult to follow at times. I would advise the authors to include more details about these works in the supplementary or even the main paper if possible.
- Related to the previous point, details about Example 3.3 in Section 3 are not provided at all.
- This is a minor suggestion, but using $x$ to denote the weights is unconventional in the machine learning community as it typically is used to denote the dataset. This may be a source of confusion to some readers.

Practicality:
- The problem setting seems impractical, as factorizing $\mathbf{A} \in \mathbb{R}^{T x T}$ is intractable in realistic settings. In the experimental settings considered in this work, T is reasonable. However, in practice $T$ can be on the order of millions and even a one-time matrix factorization is not tractable.
- Is Assumption 4.2 a realistic assumption for deep learning? The loss landscapes for neural networks are typically extremely sharp, rendering some of the convergence rate results vacuous. I understand that this is a common assumption in the optimization community.

**Questions:**

1. How can this method be scaled to large $T$?

**Limitations:**

Limitations are adequately discussed.

---

> ### Author Rebuttal · Authors · 2023-08-09
>
> ## Clarity
> 1. Due to the page limitations we unfortunately were unable to fit all the background information. Upon your suggestion, we will add more details in the appendix in the next version or in the main text if space permits. In particular we will include details of how to compute sensitivity of C, and will elaborate on Example 3.3.
>
> ## Practicality
> 1. The reviewer is correct to point out that the class of methods we consider are not currently applicable to the T >= 1M setting.  It is an interesting and important open problem to scale up the matrix factorization mechanism to these settings, but this is a problem that is orthogonal to what we study in this work.
>
> 2. The reviewer correctly points out that assumption 4.2 is a simplifying assumption that is not applicable to deep learning loss functions.  Nevertheless it is an assumption that is commonly used in the optimization literature that allows us to get a theoretical handle on the problem.  While our theory does not cover deep learning loss functions, our experiments on CIFAR-10 and StackOverflow do, and our method works well empirically in both of those settings.

---

### Official Review · Reviewer_dQeP · 2023-07-28

**Soundness:** 4 excellent
**Presentation:** 3 good
**Contribution:** 3 good
**Rating:** 7
**Confidence:** 4

**Summary:**

This paper studies variants of noisy gradient descent, focusing on when the noise added at different time steps may be dependent. The motivation of this is differentially private optimization: while the standard DP-(S)GD adds independent noise at each time step, the recent DP-FTRL mechanism [1] adds carefully correlated noise. Even more recent work [2,3] extends this approach via the matrix mechanism. This paper builds heavily on the latter two works. It begins by identifying how the prior analysis is pessimistic and settings where one might hope to improve upon it.

We then analyze in detail "perturbed gradient descent," PGD, ie gradient descent with independent Gaussian noise, and "anti-PGD," which after each gradient step we add a new noise term and *subtract* the noise term from the previous time step. This latter method is closely related to randomized smoothing.

The authors establish the main theorems (for general "noisy SGD-type" algorithms without clipping) via a novel analysis, analyzing virtual iterates that periodically revert to the actual sequence.

The authors perform experiments, first on synthetic data without gradient clipping (to validate their theorems) and then with clipping on benchmark data sets. They show that, in some settings, their approach matches or exceed DP-SGD.

[1] Kairouz, P., McMahan, B., Song, S., Thakkar, O., Thakurta, A., & Xu, Z. (2021, July). Practical and private (deep) learning without sampling or shuffling. In International Conference on Machine Learning (pp. 5213-5225). PMLR.

[2] Denisov, S., McMahan, H. B., Rush, J., Smith, A., & Guha Thakurta, A. (2022). Improved differential privacy for sgd via optimal private linear operators on adaptive streams. Advances in Neural Information Processing Systems, 35, 5910-5924.

[3] Choquette-Choo, C. A., McMahan, H. B., Rush, K., & Thakurta, A. (2022). Multi-epoch matrix factorization mechanisms for private machine learning. arXiv preprint arXiv:2211.06530.

**Strengths:**

The paper investigates a topic of clear theoretical and practical importance; there is substantial food for thought. In particular, the authors convince me that there is more to be done in this direction. They analyze simplified cases but lay out many directions for further analysis. I agree that the paper "highlights the wealth of stochastic optimization questions arising from recent advances in differentially private model training."

In addition, the paper makes clear progress beyond prior work, bringing a new set of tools to bear.

I thought the analysis for the main theorems was very cool. It seems plausible that this type of analysis might be extended or applied elsewhere.

I feel the authors do a good job placing their work within the broader literature.

**Weaknesses:**

The paper has limited scope and there were fewer conclusions after the main theorems than I had hoped. For instance, in line 253 we read "Intuitively, since $\lVert \Lambda_{\tau} \mathbf{B}\rVert_{F}^{2}$ is a better proxy for learning performance than $\lVert \mathbf{B}\rVert_{F}^2$, minimizing this quantity... should lead to... better privacy-utility tradeoffs." The "intuition" just repeats the paper's setup, that of finding better factorization. I would like a discussion of *why* this proxy is better, beyond the theorem statement.

I feel that future work may supercede this approach without building on it. It is not clear how much this paper would contribute beyond pointing out that there is more work to be done. Indeed, perhaps algorithms from from Choquette-Choo et al. already outperform this approach? While the contribution is clear, I feel there is a ceiling on its impact.

I feel the paper spends too little time on background. This is on purpose, since the setup is similar to that of Denisov et al. and Choquette-Choo et al. However, I would like to see a more self-contained discussion. Two examples in particular stand out to me: first, the expression sens(C) is never defined. Second, as far as I can tell there is no detail on how DP-SGD was run as an experimental baseline.

Although the motivation and experiments deal with privacy, the main theoretical results are for nonprivate algorithms. I was confused by this at first, especially in the transition from Section 2 to 3: the paper says "we derive a tighter bound... to design better factorization mechanisms for differentially private optimization," and then almost immediately says "we omit gradient clipping from our analysis."

**Questions:**

What is the DP-SGD you're comparing against? How were the hyperparameters selected?

Can you elaborate on what kinds of factorizations your approach favors? Do you have an intuition for how they differ from those of Denisov et al.?

The main theorems specify a value of $\tau$, but the experimental performance seems to improve monotonically with $\tau$. Maybe I missed it, but do you have an idea for why that is?

**Limitations:**

I feel the authors are clear about the limitations of their work. As I see it, the main limitations are (i) the narrow scope and (ii) the experiments, which I feel are more "illustrative" rather than comparing to the state of the art.

---

> ### Author Rebuttal · Authors · 2023-08-09
>
> We would like to thank the reviewer for their detailed review and positive assessment of our paper ! Below we address the raised questions:
>
> ## Weaknesses:
>
> 1. We write that $‖\Lambda_{\tau}  B‖^2_F$  is a better proxy for learning performance because it captures convergence of the algorithm more tightly that the $‖B ‖^2_F$. Indeed, it holds that $|| \Lambda_{\tau} B||^2_F \leq 4 ||B||^2_F$, making our rates in Theorem 4.7. strictly tighter than that in (5). Moreover, our derived convergence rate in Theorems 4.6. and 4.7. is tight for some notable special cases such as PGD, Anti-PGD and Chess-PGD (discussed in Appendix A.2), while the prior convergence rate based on $||B||^2_F$ cannot be tight for these cases (see discussion in lines 130-137, as well as in Appendix A). Therefore, we believe that minimizing a tighter bound based on $||\Lambda_{\tau} B||^2_F$ (under the fixed privacy) should directly transfer to the better optimization properties than minimizing objective based on a loose bound $||B||^2_F$. We will add this discussion to the next version.
>
>
> 2. Our work analyzes the dynamics of a general matrix mechanism applied to model training with gradient descent, and is therefore applicable to any of the matrix-based mechanisms recently designed for improved scalability. To the best of our knowledge, none of the related work has moved beyond the Frobenius-norm formulation of error including (Choquette-Choo et al.), which we show to fail to be predictive of optimization performance in a variety of cases. This is to say: our work inherits the applicability, algorithmic and scalability improvements pursued in the rest of the literature, but addresses a fundamentally orthogonal problem. For this reason, we expect the domain of application of the analysis presented here to continue to grow in the near future.
>
>     In particular, algorithms from (Choquette-Choo et al.) provide an orthogonal contribution to our work. In (Choquette-Choo et al.) the major contribution lies in extending computation of sensitivity of $C$ from having only one pass over the data (Denisov et al.) to the multi-epoch algorithms, and proposing an efficient way to solve the resulting matrix factorization optimization problem. However, their algorithms are still based on minimizing the objective $||B||$ (see equation (4) in Choquette-Choo et al. ). In our experiments in section 6 we already combined multi-epoch matrix mechanisms from (Choquette-Choo et al.) with our new proposed objective $|| \Lambda_{\tau} B||$, as we do multiple passes over the data in all the settings.
>
>
> 3. Limited by space we decided to skip some of the details that are not crucial to understanding our main contributions. Upon your suggestion, we will add more details in the appendix in the next version or in the main text if space permits.
>
> 4. By choosing a set of assumptions and relaxations in Section 3, we wanted to find the simplest model for which it would be meaningful to study the effect of the correlated noise. By choosing a simpler model we are able to study tightly the effect of the correlated noise structures. Moreover, if the function is Lipshitz, its gradients do not require clipping.
> In experiments we illustrate that our approach can lead to practical improvements even with the simplifications made.
>
> ## Questions:
>
> 1. For MNIST experiments we fixed the clipping threshold at 1.0 and the learning rate at 0.5. For the CIFAR-10 experiments we fixed the clipping threshold at 1.0 and we tuned the learning rate over a grid of {2^k for k=-4, -3, …, 3, 4} independently for each of the methods.
>
> 2. We plot the two factorizations (MF and MF+) in the attached PDF.
> Because of the repetitive pattern in $\Lambda_{\tau}$, our proposed factorization MF+ consists of exactly the same blocks of size $\tau$, with noise being correlated only within these small blocks, and not correlated otherwise. We will further polish and add it to the next version.
>
> 3. We believe that this is in part because of differences between the last-iterate, and the averaged iterate performances, as we illustrated in our experiments on a random quadratic function (Fig. 2). Specifically, but setting $\tau = T, \Lambda_{\tau}$ is a diagonal matrix where all entries are equal, except for the last which is much larger.  This influences the objective by putting much more weight on the error of the last-iterate model.  In our experiments (section 6.1) we evaluate the final trained model at the last trained model (last-iterate), while the theory gives guarantees for the averaged iterate performance. Studying last-iterate behavior theoretically is an interesting direction for future work.

---

> > ### Comment · Reviewer_dQeP · 2023-08-11
> >
> > Thanks for the detailed comments. My only comment (to which you need not reply) is that I feel your response to the first weakness misunderstands my concern. From the submission, I understood that $\lVert \Lambda_{\tau} B\rVert_F^2$ yields stronger theoretical guarantees that are tight in special cases. I was looking for more intuition, such as an informal description of what your proxy captures that $\lVert B \rVert_F^2$ does not.

---

> > > ### Author Response · Authors · 2023-08-18
> > >
> > > We would like to thank the reviewer for their prompt response !
> > > Intuitively, our convergence rate accounts only for correlations in noise that are close in terms of round numbers (within the $\tau$ closest iterations) and does not consider noise correlations for round numbers that are far apart. We will clarify this in the main text.

---

### Author Rebuttal · Authors · 2023-08-09

We would like to thank all of the reviewers for their time spent to review our paper, their useful comments that will allow to improve our paper, and for their positive assessment of our work. Below we address comments from each of the reviewers separately. We will try to clarify most of these points in the next version of the paper as well.

Upon the question of the reviewer dQeP, in the attached PDF we illustrate noise patterns $B$ found by minimizing the previous factorization objective in Problem 2.2. $\text{OPT}(A)$ (Fig. a) and our proposed objective in Section 5 $\text{OPT}(\Lambda_{\tau} A)$ (Fig. b). We see that $B_{\text{MF}}$ cancels gradually the noise added at the previous iterations throughout all of the iterations, while for $B_{\text{MF}^+}$ the noise cancels only within the last $\tau$ iterations. Moreover, $B_{\text{MF}^+}$ consists of exactly the same blocks of size $\tau$, making it more computationally efficient to solve $\text{OPT}(\Lambda_\tau A)$ when $\tau < T$.

---

### Comment · Area_Chair_R7es · 2023-08-20
**Only the smooth case?**

From a quick look it appears that the main difference from the previous analysis is that it is based on adapting existing analyses for smooth convex optimization as opposed to the general convex case (as in Denisov et al) and then optimizing the resulting matrix decomposition. If that's the case then the main claim that the achieved bounds are tighter than the previous work seems misleading as the new bound requires stronger assumptions (and thus is incomparable to prior work). Could you please clarify this point?

---

> ### Author Response · Authors · 2023-08-20
>
> Dear AC,
>
> We would like to thank you for your question. Indeed, our assumptions are different from assumptions used in the analysis of (Denisov et al): we analyze smooth convex and non-convex functions, while (Denisov et al) analyze Lipschitz strongly convex functions.
>
> However, even under assumptions of (Denisov et al), the convergence rate based on $||B||_F$ is not tight. Please note that our discussion in lines 130-134 refers to the rate from [39] that is derived under the same assumptions as in (Denisov et al). This rate from [39] for a special case of PGD noise correlation (when B = A) is tighter than the one derived in (Denisov et al).
>
> Moreover, our analysis is **not** simply based on adapting existing proof techniques of smooth optimization, as we detailed in Section 4.1. In this section, we reviewed two main existing techniques of smooth optimization: real-iterate analysis and perturbed iterate analysis, and showed that it is non trivial to obtain a tight rate using either of these two techniques. See our discussion in lines 218-220, as well as appendix F.
>
> Our analysis is novel, and is based on the *non-trivial* combination of the two existing proof techniques through utilizing virtual sequence with restarts (see Section 4.2). In particular, lines 533-541 in the non-convex proof, as well as lines 558-572 in the convex proof *are non trivial and novel* steps in the proofs.
>
> We chose to impose smoothness as (i) common in optimization literature, and (ii) because it allows us to theoretically study an important class of non-convex functions, thus being closer to the deep learning setting. In experiments we illustrate that our approach can lead to practical improvements for both: differentially private convex learning (MNIST, logistic regression) as well as deep learning training (CIFAR10, CNN). We will clarify this in the main text.
>
> To sum up, all the reviewers positively assessed our paper, with each of them commenting on the novelty and soundness of our analysis:
> - Reviewer dQeP: *“the paper makes clear progress beyond prior work, bringing a new set of tools to bear. I thought the analysis for the main theorems was very cool. It seems plausible that this type of analysis might be extended or applied elsewhere.”*
> - Reviewer ptMd: *”theoretical results are very involved”*
> - Reviewer WD4o: *“Interestingly, the proposed analysis gives a unified analysis of PGD and Anti-PGD, that are difficult to analyze together otherwise.”*
> - Reviewer 2HBd: *”The main contribution is a new convergence analysis based on the idea of "restart iterations" ”*
> - Reviewer xFGx: *” the paper is theoretically very sound ”*
>
> We would be happy to provide additional clarifications whether you have any remaining questions or concerns regarding our paper !
>
> [39] S. Shalev-Shwartz, O. Shamir, N. Srebro, and K. Sridharan. Stochastic convex optimization. In 447 Annual Conference Computational Learning Theory, 2009.

---

> > ### Comment · Area_Chair_R7es · 2023-08-21
> >
> > I do not see how the discussion in lines 130-134 addresses my point. Indeed prior bounds may be loose but your bounds require additional assumption to improve on them. Not particularly important but I do not see a strong convexity assumption in your restatement of Denisov et al.
> > To clarify I agree that smoothness is a common assumption and deriving better bounds for it is important.  But my point is that convergence is often faster under the assumptions of smoothness. So the issue is that the high-level claims of direct improvement over prior work look misleading to me (and might have mislead reviewers as well).

---

> > > ### Author Response · Authors · 2023-08-21
> > >
> > > Dear AC,
> > >
> > > We would like to thank you for your follow-up. We are sorry for the confusion caused regarding the assumptions that we make, and we thank you for pointing to that. We will make our discussions clearer in the next version of the paper.
> > >
> > > We agree that it is incorrect to directly compare the rates from (Denisov et al) and our derived convergence rates in Theorems 4.6 and 4.7. In fact, our paper never directly compares these two rates. We want to highlight that in the contributions section in lines 40-50, as well as below our main Theorems 4.6 and 4.7 we never mention that our rates are tighter than that of (Denisov et al).
> > >
> > > Instead, in section 4, we establish the two baseline approaches using existing tools of smooth optimization and identify that these two baselines are not tight (see Appendix F). Our improved convergence rate is based on novel proof techniques and clearly improves upon established baselines in section 4.1. This is what we refer to in the abstract as being “demonstrably tighter than prior work”. We will further clarify this in the paper.
> > >
> > > It is worth noting that the analysis in (Denisov et al) is based on perturbed iterate analysis, but applied to the non-smooth optimization setting. In the smooth setting, perturbed iterate analysis gives the rate based on $||B||_F$, similarly to the non-smooth in (Denisov et al). In section 4.1. we established (for the smooth setting) that this rate based on $||B||_F$ is tight for Anti-PGD but not for PGD, and also showed a lower bound that perturbed iterate analysis *cannot be tight* for smooth optimization (see Appendix F).
> > >
> > > In conclusion, we first thank the AC for pointing out the inaccuracy in our discussions, we will make our discussion clearer in the next version.
> > > Despite using different assumptions from (Denisov et al), we carefully set the baselines for the assumptions that we used. We therefore believe that our paper makes a non-trivial contribution to the smooth optimization literature bringing a new set of analytic tools. We hope that our answer could resolve your concern and we are happy to discuss this further.

---

> > > > ### Comment · Reviewer_dQeP · 2023-08-21
> > > >
> > > > I thank the AC for identifying this issue. I had not understood that the assumptions differed between this work and Denisov et al., and was under the impression that the two operated in the same setting. For instance, the authors' rebuttal to me says "making our rates in Theorem 4.7. strictly tighter than that in (5)", where (5) is the equation with Denisov et al.'s rates. Of course, I should have read more closely.
> > > >
> > > > Looking over the submission and the discussion here, I think part of the issue is the word "tighter," which is used in several places where it might be more appropriate to say "faster."
> > > >
> > > > I feel that (i) I was at least slightly misled by the high-level claims of direct improvement over prior work, and (ii) analyzing the smooth setting is still important.

---

### Decision · Program_Chairs · 2023-09-21

**Decision:**

Accept (poster)

**Comment:**

This paper studies variants of differentially private gradient descent with noise added at different time steps may be dependent. It gives a new analysis for the smooth case that is, in certain cases, tighter than the one known for the non-smooth case. The analysis relies on a new combination of existing analyses and small scale experiments show a benefit of the resulting algorithm.